# Learning to Annotate Part Segmentation with Gradient Matching

**Yu Yang**
Department of Automation
Tsinghua University, BNRist
yang-yu16@mails.tsinghua.edu.cn

**Xiaotian Cheng**
Department of Automation
Tsinghua University, BNRist
cxt20@mails.tsinghua.edu.cn

**Hakan Bilen**
School of Informatics
University of Edinburgh
hbilen@ed.ac.uk

**Xiangyang Ji**
Department of Automation
Tsinghua University, BNRist
xyji@tsinghua.edu.cn

## Abstract

The success of state-of-the-art deep neural networks heavily relies on the presence of large-scale labelled datasets, which are extremely expensive and time-consuming to annotate. This paper focuses on tackling semi-supervised part segmentation tasks by generating high-quality images with a pre-trained GAN and labelling the generated images with an automatic annotator. In particular, we formulate the annotator learning as a learning-to-learn problem. Given a pre-trained GAN, the annotator learns to label object parts in a set of randomly generated images such that a part segmentation model trained on these synthetic images with their predicted labels obtains low segmentation error on a small validation set of manually labelled images. We further reduce this nested-loop optimization problem to a simple gradient matching problem and efficiently solve it with an iterative algorithm. We show that our method can learn annotators from a broad range of labelled images including real images, generated images, and even analytically rendered images. Our method is evaluated with semi-supervised part segmentation tasks and significantly outperforms other semi-supervised competitors when the amount of labelled examples is extremely limited.

## 1 Introduction

In recent years, deep neural networks have shown a remarkable ability to learn complex visual concepts from large quantities of labelled data. However, collecting manual annotations for a growing body of visual concepts remains a costly practice, especially for pixel-wise prediction tasks such as semantic segmentation. A common and effective way of reducing the dependence on manual labels is first learning representations from unlabeled data (Pathak et al., 2016; Noroozi & Favaro, 2016; Gidaris et al., 2018; He et al., 2020) and then transferring learned representations to a supervised learning task (Chen et al., 2020; Zhai et al., 2019). The insight of this paradigm is that some of the transferred representations are informative, easily identified, and disentangled, such that a small amount of labelled data suffices to train the target task.

A promising direction for learning such representations is using generative models (Kingma & Welling, 2014; Goodfellow et al., 2014) since they attain the capability of synthesizing photorealistic images and disentangling factors of variation in the dataset. Recent methods (Goetschalckx et al., 2019; Shen et al., 2020; Karras et al., 2019; Plumerault et al., 2020; Jahanian et al., 2020; Voynov & Babenko, 2020; Spingarn-Eliezer et al., 2021; Härkönen et al., 2020) show that certain factors such as object shape and position in the images synthesized by generative adversarial networks (GANs) (Goodfellow et al., 2014) can be individually controlled by manipulating latent features. Building on the success of powerful GAN models, Zhang et al. (2021) and Tritrong et al. (2021) demonstrate that feature maps in StyleGAN (Karras et al., 2019; 2020b;a) can be mapped to semantic segmentation masks or keypoint heatmaps through a shallow decoder, denoted as *annotator* in this

paper. Remarkably, the annotator can be trained after a few synthesized images are manually labelled, and the joint generator and trained annotator can be used as an infinite labelled data generator. Nonetheless, labelling synthesized images with "human in the loop" has two shortcomings: (i) it can be difficult to label the generated images of low quality; (ii) it requires continual human effort whenever the generator is changed.

One straightforward way to mitigate these limitations is to inverse the generation process to obtain generator features for given images. In this way, annotator, which only receives generator features as input, can be trained from existing labelled data, not necessarily generated data. However, inversion methods can not ensure exact recovery of the generator features, leading to degraded performance of annotator (see Section C.1). Another way to address this issue is to align the joint distribution of generated images and labels with that of real ones using adversarial learning as in SemanticGAN (Li et al., 2021). However, adversarial learning is notoriously unstable (Brock et al., 2018) and requires a large amount of data to prevent discriminators from overfitting (Karras et al., 2020a).

In this paper, we formulate the annotator learning as a learning-to-learn problem – the annotator learns to label a set of randomly generated images such that a segmentation network trained on these automatically labelled images obtains low prediction error on a small validation set of manually labelled images. This problem involves solving a nested-loop optimization, where the inner loop optimizes the segmentation network and the outer loop optimizes the annotator based on the solution of inner-loop optimization. Instead of directly using end-to-end gradient-based meta-learning techniques (Li et al., 2019; Pham et al., 2021), we reduce this complex and expensive optimization problem into a simple gradient matching problem and efficiently solve it with an iterative algorithm. We show that our method obtains performance comparable to the supervised learning counterparts (Zhang et al., 2021; Tritrong et al., 2021) yet overcomes their shortcomings such that training annotators can utilize labelled data within a broader range, including real data, synthetic data, and even out-of-domain data.

Our method requires a large quantity of unlabeled data for pre-training GANs and a relatively small amount of labelled data for training annotators, which drops into a semi-supervised learning setting. Our method relies on unconditional GANs, of which the performance limit the application of our approach. As the state-of-the-art GANs only produce appealing results on single-class images but struggle to model complex scenes, we focus on part segmentation of particular classes and avoid multi-object scenes.

Our contribution can be summarized as follows. (i) We formulate the learning of annotations for GAN-generated images as a learning-to-learn problem and propose an algorithm based on gradient matching to solve it. Consequently, a broad range of labelled data, including real data, synthetic data, and even out-of-domain data, is applicable. (ii) We empirically show that our method significantly outperforms other semi-supervised segmentation methods in the few-label regime.[1]

## 2 RELATED WORK

**Semi-supervised learning** Semi-supervised learning (SSL) (Zhu, 2005) augments the training of neural networks from a small amount of labelled data with a large-scale unlabeled dataset. A rich body of work regularizes networks on unlabeled data with consistency regularization. In particular, networks are required to make consistent predictions over previous training iterations (Laine & Aila, 2017; Tarvainen & Valpola, 2017; Izmailov et al., 2018), noisy inputs (Miyato et al., 2018), or augmented inputs (Berthelot et al., 2019b;a; Sohn et al., 2020). This paradigm is also interpreted as a teacher-and-student framework where the teacher produces pseudo labels for unlabelled data to supervise the training of students. Pseudo labels can be enhanced with heuristics (Lee et al., 2013; Berthelot et al., 2019a) or optimized towards better generalization (Pham et al., 2021; Li et al., 2019). Apart from image classification tasks, the above ideas are also successfully applied to semantic segmentation tasks (Hung et al., 2018; Ouali et al., 2020; Ke et al., 2020; French et al., 2020). Our work can also be interpreted from a teacher-and-student perspective. In contrast to the above methods that adopt homogeneous network structures for teacher and student, our work employs generative models as a teacher and discriminative models as a student.

Other SSL approaches explore the utility of unlabelled data from a representation learning perspective. In particular, deep neural networks are first optimized on the unlabeled data with self-supervised

---

[1]Code is available at https://github.com/yangyu12/lagm.

learning tasks (Pathak et al., 2016; Noroozi & Favaro, 2016; Gidaris et al., 2018; He et al., 2020) to gain powerful and versatile representations, and then finetuned on the labelled data to perform target tasks (Chen et al., 2020; Zhai et al., 2019). In addition, generative models can also learn efficient and transferable representations without labels. To this end, one promising direction is to decode generative features into dense annotations such as segmentation masks and keypoint heatmap to facilitate SSL, as done in SemanticGAN (Li et al., 2021), DatasetGAN (Zhang et al., 2021), RepurposeGAN (Tritrong et al., 2021) and our work. However, our work differs from these works in that we formulate the annotator learning as a learning-to-learn problem that is solved with gradient matching. Particularly, unlike SemanticGAN (Li et al., 2021) that learns annotator and data generator jointly, our method presumes a fixed pre-trained GAN and learns annotator only, which circumvents the complication of joint learning.

**Semantic Part Segmentation** Semantic part segmentation, which decomposes rigid or non-rigid objects into several parts, is of great significance in tremendous computer-vision tasks such as human parsing (Dong et al., 2014; Nie et al., 2018; Fang et al., 2018; Gong et al., 2017), pose estimation (Yang & Ramanan, 2011; Shotton et al., 2011; Xia et al., 2017) and 3D object understanding (Yi et al., 2016; Song et al., 2017). The challenge of part segmentation arises from viewpoint changes, occlusion, and a lack of clear boundaries for certain parts. Existing datasets manage to alleviate the problem by enriching the data and annotations. CelebAMask-HQ (Lee et al., 2020a) is a large-scale face image dataset annotated with masks of face components. Chen et al. (2014) provides additional part segmentation upon PASCAL VOC dataset (Everingham et al., 2010). Those datasets meet the demands of latest data-driven and deep neural network based method (Lee et al., 2020b; Wang et al., 2015; Wang & Yuille, 2015). However, a well-curated part segmentation dataset containing fine-grained pixel-wise annotation usually requires time-consuming manual labelling work, which holds back the development of this field. This problem also motivates the latest work (Zhang et al., 2021; Li et al., 2021; Tritrong et al., 2021) to leverage a generative model as an infinite data generator in a semi-supervised learning setting, which gains promising results.

**Gradient matching** Gradient matching is an important technique in meta learning (Li et al., 2018; Sariyildiz & Cinbis, 2019), which is shown to be effective in domain generalization (Li et al., 2018), zero-shot classification (Sariyildiz & Cinbis, 2019), and dataset condensation (Zhao et al., 2021; Zhao & Bilen, 2021) *etc*. Although our work's general principle and gradient matching loss function resemble those in the above works, we are motivated to solve a different problem. In particular, Gradient Matching Network (GMN) (Sariyildiz & Cinbis, 2019) employs gradient matching to train a conditional generative model towards application on zero-shot classification. Dataset Condensation (DC) (Zhao et al., 2021; Zhao & Bilen, 2021) condenses a large-scale dataset into a small synthetic one with gradient matching. Our work shows that gradient matching is also an effective way to learn annotations of GAN-generated images from limited labelled examples. Moreover, in contrast to DC, where the synthesized images are typically not realistic, our work is concerned about photo-realistic synthetic images.

## 3 METHOD

### 3.1 PRELIMINARY

Let $G$ denote a generative model trained from a dataset $\mathcal{D}_{tot}$ without labels. $G$ can be trained via adversarial learning (Goodfellow et al., 2014) such that it can generate photo-realistic images. The generation process is formulated as

$$(\mathbf{h}, \mathbf{x}) = G(\mathbf{z}), \quad \mathbf{z} \sim P_z \qquad (1)$$

where $\mathbf{z}$ denotes a random variable drawn from a prior distribution $P_z$ which is typically a normal distribution, $\mathbf{x}$ denotes the generated image, and $\mathbf{h}$ represents the hidden features in $G$. An annotator $A_\omega$ is constructed to decode $\mathbf{h}$ into labels

$$\hat{\mathbf{y}} = A_\omega(\mathbf{h}), \qquad (2)$$

where $\omega$ parameterizes the annotator. Let $\mathcal{D}_l$ denote a manually labeled dataset. Given a fixed $G$, our problem is to learn annotator $A_\omega$ from $\mathcal{D}_l$ such that $A_\omega$ can annotate images generated from $G$.

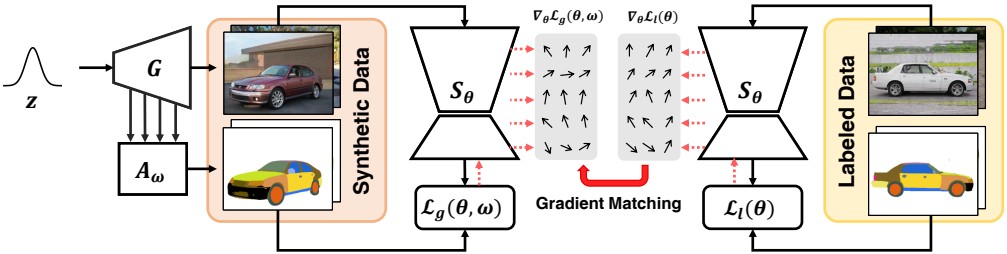

Figure 1: **Illustration** of learning to annotate with gradient matching. The procedure is as follows. (i) The gradients of segmentation network $S_\theta$ on labeled examples are computed, denoted as $\nabla_\theta \mathcal{L}_l(\theta)$. (ii) A batch of synthetic data is *randomly* generated by generator $G$ and labeled by annotator $A_\omega$. (iii) The gradients of $S_\theta$ on synthetic data are computed, denoted as $\nabla_\theta \mathcal{L}_g(\theta, \omega)$. (iv) Gradient matching between $\nabla_\theta \mathcal{L}_l(\theta)$ and $\nabla_\theta \mathcal{L}_g(\theta, \omega)$ is computed to optimize $A_\omega$.

**Revisiting supervised annotator learning**  DatasetGAN (Zhang et al., 2021) and Repur-poseGAN (Tritrong et al., 2021) train annotators in a supervised learning manner. First, a set of synthetic images $\{\mathbf{x}_i\}_{i=1}^N$ generated from $G$ are selected and annotated by humans and their generator features $\{\mathbf{h}_i\}_{i=1}^N$ are reserved. These features and manual labels $\{\mathbf{y}_i\}_{i=1}^N$ constitute the labeled dataset $\mathcal{D}_l = \{(\mathbf{x}_i, \mathbf{h}_i, \mathbf{y}_i)\}_{i=1}^N$. Second, an annotator is trained by minimizing the loss computed between annotator prediction and ground truth labels,

$$\omega = \arg \min_\omega \mathbb{E}_{(\cdot, \mathbf{h}, \mathbf{y}) \sim \mathcal{D}_l} f(A_\omega(\mathbf{h}), \mathbf{y}), \qquad (3)$$

where $f$ denotes per-pixel cross-entropy function for segmentation tasks. This method, however, requires $\mathcal{D}_l$ to be a *synthetic* and *generator-specific* set, which has the following two defects. (i) Details might be illegible in synthetic images, which increases the difficulty of annotation. (ii) Data needs to be re-labelled whenever the generator is changed. Although projecting labelled images into GAN latent space can alleviate the above drawbacks, it leads to degraded performance due to inexact inversion (see Section C.1 in the appendix).

## 3.2 LEARNING TO ANNOTATE WITH GRADIENT MATCHING

**Learning to learn**  We formulate the problem of learning $A_\omega$ as a learning-to-learn problem like Li et al. (2019); Pham et al. (2021). In particular, we introduce a segmentation network $S_\theta$, a deep neural network parameterized with $\theta$. It takes as input an image and outputs a segmentation mask. This segmentation network learns the target task only from the automatically labeled generated images, which is formulated as

$$\theta^*(\omega) = \arg \min_\theta \underbrace{\mathbb{E}_{(\mathbf{h}, \mathbf{x})=G(\mathbf{z}), \mathbf{z} \sim P_z} f(S_\theta(\mathbf{x}), A_\omega(\mathbf{h}))}_{\mathcal{L}_g(\theta, \omega)}. \qquad (4)$$

where $f$ denotes the loss function for the target task. The optimal solution of Equation 4 depends on the $\omega$, which is therefore denoted as $\theta^*(\omega)$. The performance of learned segmentation network can be evaluated with loss on the labeled dataset $\mathcal{D}_l$ as

$$\mathcal{L}_l(\theta^*(\omega)) = \mathbb{E}_{(\mathbf{x}, \mathbf{y}) \in \mathcal{D}_l} f(S_{\theta^*(\omega)}(\mathbf{x}), \mathbf{y}). \qquad (5)$$

The aim of learning the annotator is to produce high-quality labels such that the learned segmentation network achieve minimal loss on $\mathcal{D}_l$, which is formulated as

$$\begin{aligned} \min_\omega \quad & \mathcal{L}_l(\theta^*(\omega)), \\ \text{where} \quad & \theta^*(\omega) = \arg \min_\theta \mathcal{L}_g(\theta, \omega). \end{aligned} \qquad (6)$$

This problem is a nested-loop optimization problem, where the inner loop optimizes the segmentation network (Equation 4) and the outer loop optimizes the annotator based on the solution of the inner loop. This optimization problem does not require manual labelling of generated images. To this end, any labelled images, including *real* labelled images, can serve as $\mathcal{D}_l$. While this problem can be solved with MAML-like (Finn et al., 2017) approaches (Li et al., 2019; Pham et al., 2021), its solution involves expensive unrolling of the computational graph over multiple $\theta$ states, which is memory and computation intensive. Hence, we reduce it into a simple gradient matching problem as follows. See Section C.1 in the appendix for further comparison and discussion.

**Gradient matching**   Inspired by Li et al. (2018), we reduce the nested-loop optimization problem (Equation 6) into a gradient matching problem. First, as commonly done in gradient-based meta learning approaches, the optimization of segmentation network (Equation 4) is approximated with one-step gradient descent,

$$\theta^*(\omega) \approx \theta_0 - \eta\nabla_\theta\mathcal{L}_g(\theta_0, \omega), \tag{7}$$

where $\eta$ denotes the learning rate and $\theta_0$ denotes the initialized parameter. By plugging this equation into $\mathcal{L}_l(\theta^*(\omega))$, unfolding it as Taylor series and omitting the higher order term, we have

$$\begin{aligned}
\mathcal{L}_l(\theta^*(\omega)) &\approx \mathcal{L}_l(\theta_0 - \eta\nabla_\theta\mathcal{L}_g(\theta_0, \omega)) \\
&= \mathcal{L}_l(\theta_0) - \eta\nabla_\theta^\top\mathcal{L}_l(\theta_0)\nabla_\theta\mathcal{L}_g(\theta_0, \omega) + \dots \\
&\approx \mathcal{L}_l(\theta_0) - \eta\nabla_\theta^\top\mathcal{L}_l(\theta_0)\nabla_\theta\mathcal{L}_g(\theta_0, \omega)
\end{aligned} \tag{8}$$

The last approximation is conditioned on $\|\eta\nabla_\theta\mathcal{L}_g(\theta_0, \omega)\| \leq \epsilon$, where $\epsilon$ is a very small amount. By further omitting the constant term $\mathcal{L}_l(\theta_0)$ in Equation 8, we obtain the approximation of Equation 6 as

$$\min_\omega -\nabla_\theta^\top\mathcal{L}_l(\theta_0)\nabla_\theta\mathcal{L}_g(\theta_0, \omega) \qquad s.t. \ \|\eta\nabla_\theta\mathcal{L}_g(\theta_0, \omega)\| \leq \epsilon. \tag{9}$$

We notice a connection between this constrained optimization problem with the following unconstrained optimization problem

$$\min_\omega -\frac{\nabla_\theta^\top\mathcal{L}_l(\theta_0)\nabla_\theta\mathcal{L}_g(\theta_0, \omega)}{\|\eta\nabla_\theta\mathcal{L}_g(\theta_0, \omega)\|}, \tag{10}$$

where to minimize this objective, one needs to simultaneously maximize the dot product between the gradients and minimize the norm of $\nabla_\theta\mathcal{L}_g(\theta_0, \omega)$. Note that $-\frac{\nabla_\theta^\top\mathcal{L}_l(\theta_0)\nabla_\theta\mathcal{L}_g(\theta_0, \omega)}{\|\eta\nabla_\theta\mathcal{L}_g(\theta_0, \omega)\|} = \frac{\|\nabla_\theta\mathcal{L}_l(\theta_0)\|}{\eta}(1 - \frac{\nabla_\theta^\top\mathcal{L}_l(\theta_0)\nabla_\theta\mathcal{L}_g(\theta_0, \omega)}{\|\nabla_\theta\mathcal{L}_l(\theta_0)\|\|\nabla_\theta\mathcal{L}_g(\theta_0, \omega)\|}) - \frac{\|\nabla_\theta\mathcal{L}_l(\theta_0)\|}{\eta}$, and the constant term and constant coefficient can be omitted without changing the optimal solution. We re-write the learning objective of Equation 10 as

$$\mathcal{L}_{gm}(\theta_0, \omega) = D(\nabla_\theta\mathcal{L}_l(\theta_0), \nabla_\theta\mathcal{L}_g(\theta_0, \omega)) = 1 - \frac{\nabla_\theta^\top\mathcal{L}_l(\theta_0)\nabla_\theta\mathcal{L}_g(\theta_0, \omega)}{\|\nabla_\theta\mathcal{L}_l(\theta_0)\|\|\nabla_\theta\mathcal{L}_g(\theta_0, \omega)\|}, \tag{11}$$

which computes the cosine distance between the gradients of segmentation network on *automatically* labeled generated data and *manually* labeled data (see Section B in appendix for dealing with gradient matching in multi-layer neural networks). To this end, the nested-loop optimization (Equation 6) is reduced into a simple one that minimizes the gradient matching loss (Equation 11). An illustration of this procedure is presented in Figure 1.

**Algorithm**   Note that the gradient matching problem learns an annotator for a particular network parameterized with $\theta_0$. However, a desirable annotator should produce universal automatic labels for training *any* segmentation network with previously unseen parameters. Hence, we present an alternate training algorithm of annotator and segmentation network in Algorithm 1, where the segmentation network is iteratively updated using the automatically labelled synthetic data, and the annotator is iteratively updated with gradient matching. This algorithm allows the annotator to be optimized over various segmentation network states, leading to a segmentation-network-agnostic annotator. The gradient matching problem for each segmentation network parameter is solved with $K$-step gradient descent, and typically $K = 1$ works well in practice. Implementation details are available in the appendix (see Section B).

## 4   EXPERIMENTS

### 4.1   SETUP

**Datasets**   We evaluate our method on six part segmentation datasets: **CelebA**, **Pascal-Horse**, **Pascal-Aeroplane**, **Car-20**, **Cat-16**, and **Face-34**. CelebA (Liu et al., 2015) is a large-scale human face dataset where 30,000 images are annotated with up to 19 part classes, also known as CelebAMask-HQ (Lee et al., 2020a). We consider a subset of 8 face classes and use the first 28,000 images as an unlabeled set, the last 500 images as a test set, and the rest 1500 images as a labelled set, as in Li et al. (2021). Pascal-Horse and Pascal-Aeroplane are constructed by taking images of horse and aeroplane from PascalPart (Chen et al., 2014) which provides detailed part segmentation annotations

---

**Algorithm 1:** Learning to annotate with gradient matching.

**Inputs :**

| | |
|---|---|
| $G$ | trained generator |
| $\mathcal{D}_l$ | set of labeled examples |
| $\omega, A_\omega, \eta_a$ | initial annotator parameters, annotator, and learning rate for annotator |
| $\theta, S_\theta, \eta_s$ | initial segmentor parameters, segmentor, and learning rate for segmentor |
| $T, K$, and $B$ | total number of optimization steps, interval of updating segmentor, and batch size |

1  **for** $t \leftarrow 1$ **to** $T$ **do**
    // update annotator $A_\omega$
2      $\mathcal{B}_l \leftarrow \{(\mathbf{x}_i, \mathbf{y}_i) \sim \mathcal{D}_l\}_{i=1}^B$         // sample a batch of labeled examples
3      $\nabla_\theta \mathcal{L}_l \leftarrow \frac{1}{B} \sum_{i=1}^B \nabla_\theta f(S_\theta(\mathbf{x}_i), \mathbf{y}_i)$         // compute gradients on $\mathcal{B}_l$
4      $\mathcal{B}_g \leftarrow \{(\mathbf{x}_j, A_\omega(\mathbf{h}_j))\}_{j=1}^B$         // sample a batch of synthetic data
5      $\nabla_\theta \mathcal{L}_g \leftarrow \frac{1}{B} \sum_{j=1}^B \nabla_\theta f(S_\theta(\mathbf{x}_j), A_\omega(\mathbf{h}_j))$         // compute gradients on $\mathcal{B}_g$
6      $\mathcal{L}_{gm} \leftarrow D(\nabla_\theta \mathcal{L}_g, \nabla_\theta \mathcal{L}_l)$         // compute gradient matching loss
7      $\omega \leftarrow \omega - \eta_a \nabla_\omega \mathcal{L}_{gm}$         // update annotator parameters

    // update segmentor $S_\theta$
8      **if** $t \mod K = 0$ **then**
9          $\mathcal{B}_g \leftarrow \{(\mathbf{x}_j, A_\omega(\mathbf{h}_j))\}_{j=1}^B$         // sample a batch of synthetic data
10         $\mathcal{L}_g \leftarrow \frac{1}{B} \sum_{j=1}^B f(S_\theta(\mathbf{x}_j), A_\omega(\mathbf{h}_j))$         // compute loss on synthetic data
11         $\theta \leftarrow \theta - \eta_s \nabla_\theta \mathcal{L}_g$         // update segmentor parameters
12     **end**
13 **end**

**Output :** annotator $A_\omega$ and segmentor $S_\theta$

---

for images from Pascal VOC 2010 (Everingham et al.). The selected images are cropped according to bounding box annotations. Full process details are available in the appendix. We finally obtain 180 training images, 34 validation images, and 225 test images to constitute Pascal-Horse, 180 training images, 78 validation images, and 269 test images to constitute Pascal-Aeroplane. In Pascal-Horse and Pascal-Aeroplane, additional images from LSUN (Yu et al., 2015) are utilized as an unlabeled set. Car-20, Cat-16, and Face-34, released by Zhang et al. (2021), are three part-segmentation datasets annotated with 20, 16, and 34 part classes for car, cat, and human face, respectively. We use the same datasets to make a fair comparison to DatasetGAN (Zhang et al., 2021). In these datasets, all the training images are images generated from pre-trained StyleGAN (Karras et al., 2019) while all the test images are real images. Car-20 contains 16 training images and 10 test images; Cat-16 contains 30 training images and 20 test images; Face-34 contains 16 training images and 20 test images.

**Pre-trained generators** We use models of StyleGAN family (Karras et al., 2019; 2020b;a) that are either trained by ourselves or publicly available. Details are provided in the appendix (see Section B).

**Evaluation** We use mean intersection over union (mIoU) to evaluate the performance of segmentation networks. On CelebA, Pascal-Horse, and Pascal-Aeroplane, we use the validation set to select checkpoints and report mIoU across all foreground classes, denoted as "FG-mIoU", on the test set. On Car-20, Cat-16 and Face-34, we follow the setting of DatasetGAN (Zhang et al., 2021) and report the cross-validation mIoU across all classes, including background on the test set.

## 4.2 SEMI-SUPERVISED PART SEGMENTATION

Since our method has a mild requirement for the labelled data, in this section, we show the effectiveness of our method under three different circumstances: real images as labelled data, synthetic images as labelled data, and out-of-domain images as labelled data.

**Real images as labeled data** Following SemanticGAN (Li et al., 2021), our methods are compared against SSL methods that have codes available[2], including Mean Teacher (MT) (Tarvainen & Valpola,

---

[2]We adopt the implementation in `https://github.com/ZHKKKe/PixelSSL`. For these SSL methods, the *real* unlabeled images are used as unlabeled set.

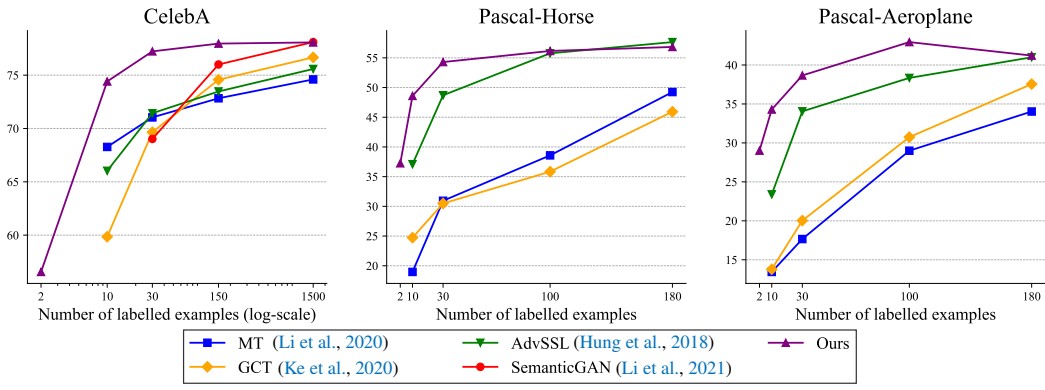

Figure 2: **Benchmark** on CelebA (*Left*), Pascal-Horse (*Middle*), and Pascal-Aeroplane (*Right*). The y-axes are FG-mIoU (%) on test set.

| Methods | $G$ | $S$ | Cat-16 # annotations: 30 # classes: 16 | Face-34 # annotations: 16 # classes: 16 | Car-20 # annotations: 16 # classes: 20 |
|---|---|---|---|---|---|
| TL[†] | - | DeepLabv3 | $21.58 \pm 0.61$ | $45.77 \pm 1.51$ | $33.91 \pm 0.57$ |
| SSL (Mittal et al., 2019)[†] | - | DeepLabv3 | $24.85 \pm 0.35$ | $48.17 \pm 0.66$ | $44.51 \pm 0.94$ |
| DatasetGAN | StyleGAN | DeepLabv3[‡] | $32.63 \pm 0.68$ | $54.55 \pm 0.25$ | $67.53 \pm 2.58$ |
| (Zhang et al., 2021) | StyleGAN | U-Net[♯] | $31.36 \pm 0.76$ | $53.84 \pm 0.41$ | $66.27 \pm 2.75$ |
| Ours | StyleGAN | DeepLabv3 | $33.89 \pm 0.43$ | $52.58 \pm 0.61$ | $63.55 \pm 2.25$ |
| | StyleGAN | U-Net | $32.64 \pm 0.74$ | $53.69 \pm 0.54$ | $60.45 \pm 2.42$ |
| | StyleGAN2 | DeepLabv3 | $33.56 \pm 0.17$ | $55.10 \pm 0.39$ | $61.21 \pm 2.07$ |
| | StyleGAN2 | U-Net | $31.90 \pm 0.75$ | $53.58 \pm 0.45$ | $58.30 \pm 2.64$ |

| | Our downstream segmentation performance | | | | | |
|---|---|---|---|---|---|---|
| | Cat-16 | | Face-34 | | Car-20 | |
| Source \ Downstream | DeepLabv3 | U-Net | DeepLabv3 | U-Net | DeepLabv3 | U-Net |
| DeepLab | $33.38 \pm 0.66$ | $33.39 \pm 0.74$ | $55.11 \pm 0.63$ | $54.77 \pm 0.32$ | $63.47 \pm 2.33$ | $62.72 \pm 2.89$ |
| U-Net | $33.38 \pm 0.40$ | $32.42 \pm 0.62$ | $54.05 \pm 0.40$ | $53.80 \pm 1.06$ | $63.22 \pm 2.42$ | $62.25 \pm 2.77$ |

Table 1: **Comparisons to DatasetGAN** on Car-20, Cat-16, Face-34. The performance is evaluated with mIoU(%). TL: transfer learning. SSL: semi-supervised learning. †: Results taken from Zhang et al. (2021). ‡: Up-to-date performance from DatasetGAN github repository. ♯: Results obtained by ourselves using DatasetGAN source codes.

2017; Li et al., 2020), Guided Collaborative Training (GCT) (Ke et al., 2020), an adversarial-learning-based semi-supervised segmentation method (AdvSSL) (Hung et al., 2018), as well as SemanticGAN (Li et al., 2021). We benchmark performances on CelebA, Pascal-Horse, and Pascal-Aeroplane with respect to a different amount of labelled data and present the results in Figure 2. Qualitative results are available in the appendix (see Figure D.7, D.8, D.9). Our method significantly outperforms other SSL methods when labelled data is extremely limited. When the number of labelled images decreases, the performance of our method drops mildly while the performances of other methods decrease drastically. We attribute it to our utilization of the highly interpretable generator features. Notably, SemanticGAN (Li et al., 2021) also exploits the generative features to produce segmentation masks, but its performance degrades faster than our methods as the number of labelled images decreases. We conjecture it is due to that the adversarial learning employed by SemanticGAN typically requires sufficient data to prevent discriminators from overfitting. The edges of our method become marginal under the case of a large number of labels. One possible reason is that the quality of pre-trained GAN needs to be further improved, which requires more research in the future. (see Section C.5 for more discussion)

**Synthetic images as labeled data** We comprehensively compare our method to a supervised learning counterpart, DatasetGAN (Zhang et al., 2021) on Car-20, Cat-16, and Face-34. We follow

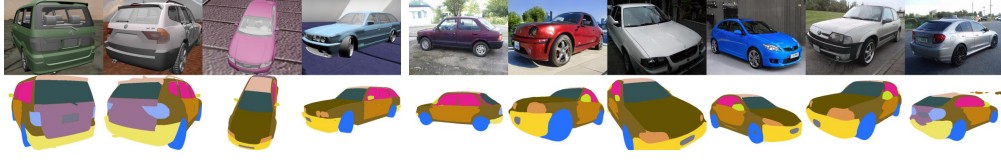

**(a) Out-of-domain labeled examples**  **(b) Generated images with learned annotations**

Figure 3: **Cross-domain annotator learning demo.** (a) Out-of-domain labelled examples: car images and ground truth segmentations rendered from 3D CAD model. (b) Generated images with learned annotations: our method learns a reasonable annotator for GAN generated images.

the same setting as in DatasetGAN, using precisely the same synthetic images as training data. Quantitative results are presented in Table 1 and qualitative comparison is available in the appendix (see Figure D.12, D.11, D.10). First, our method achieves the same performance level with Dataset-GAN on Face-34 and Cat-16. Second, as a merit of our method, we can upgrade the generator to StyleGAN2 (Karras et al., 2020b;a) without requiring extra human effort. However, upgrading the generator brings modest improvement on Face-34 and Cat-16 and even a bit decrease on Car-20. This result suggests that synthesis quality measured by metrics like FID can not precisely depict the disentanglement of generator features. Finally, our method could adapt to different segmentation architectures such as DeepLabv3 (Chen et al., 2017) and U-Net (Ronneberger et al., 2015).

We further use our trained annotator to generate synthetic segmentation datasets for training downstream segmentation networks. Table 1 shows the results when we use different source networks for training annotators with gradient matching and different downstream networks. First, our learned synthetic datasets can be used to train downstream segmentation networks of different architecture that achieve good performances. This result indicates that our learned synthetic dataset is not architecture-specific. Second, DeepLabv3 as a source network generally leads to higher performances than U-Net, suggesting that the segmentation networks used for gradient matching affect annotator learning in our method.

**Out-of-domain images as labeled data**   We further consider a challenge where out-of-domain images constitute labelled data. In particular, we render a set of car images from annotated 3D CAD models provided by CGPart[3] (Liu et al., 2021) using graphic tools (*e.g.* blender). Since 3D models are annotated with parts, the ground truth part segmentation of these rendered images are available via 3D projection. These rendered images vary in viewpoints and textures and have a significant domain gap compared to realistic car images due to inexact 3D models, unrealistic textures, artificial lighting *etc.*, raising challenges for learning annotators. Despite so, as presented in Figure 3, our method still learns reasonable annotators that produce quality segmentation labels for GAN generated images. It is noteworthy that it is nearly impossible for DatasetGAN (Zhang et al., 2021) and RepurposeGAN (Tritrong et al., 2021) to utilize such out-of-domain labelled data.

## 4.3   Ablation Study

**The frequency of updating segmentation network**   The results of an ablation study w.r.t. the interval of updating segmentation network, $K$, is shown in Figure 4. Experiments are run on Face-34 and Cat-16 with DeepLabv3 as segmentation network and StyleGAN as generator such that pairs of generator features and ground truth segmentation are available to measure the performance of the annotator quantitatively. It shows that the annotator learns significantly faster when the segmentation network is updated more frequently. The visualization shows that very detailed parts, *e.g.* pupils in Face-34 and eyes in Cat-16, emerge sooner if the segmentation network is updated more frequently. It is more effective to match gradients for various segmentation network parameters than optimize gradient matching for single segmentation network parameters.

**Gradient matching on partial segmentation network parameters**   As the implementation of gradient matching requires maintenance of computation graph of backpropagation through segmentation networks, one may be concerned about the training efficiency for very deep neural networks. Here

---

[3]https://qliu24.github.io/cgpart/

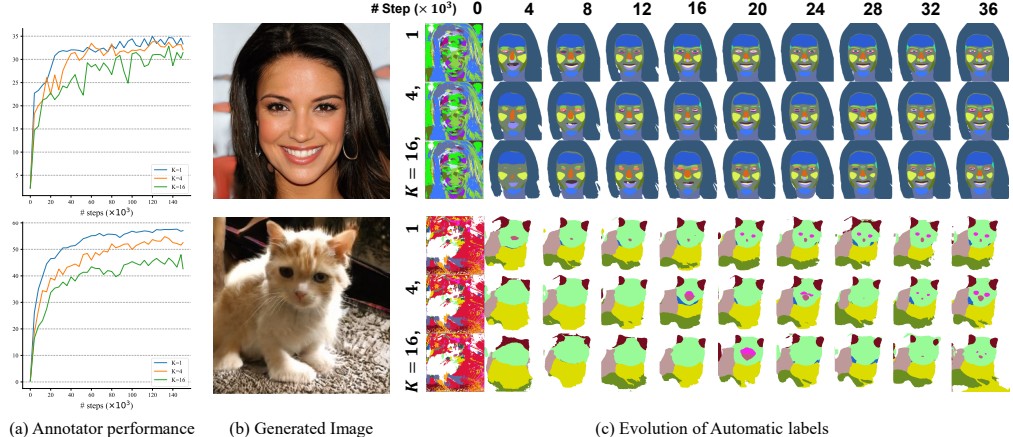

Figure 4: **Ablation study w.r.t.** $K$. Evolution of the learned annotator on Face-34 (first row) and Cat-16 (second row). (a) Annotator performance (mIoU (%)) on training data w.r.t. number of updating annotator steps ($K$). (b) Examples of generated images. (c) Evolution of automatic labels.

| Blocks | All | Res2∼Head | Res3∼Head | Res4∼Head | Res5∼Head | * Head |
|---|---|---|---|---|---|---|
| Time / step | 3.34× | 3.34× | 3.28× | 3.08× | 1.62× | 1× |
| mIoU (%) | 33.71 ± 0.77 | 34.38 ± 0.49 | 33.81 ± 0.99 | 33.78 ± 0.70 | 34.12 ± 0.57 | 33.56 ± 0.17 |

Table 2: **Ablation study w.r.t. blocks for matching gradients**. Evaluation is performed on Cat-16, where StyleGAN2 is the generator and DeepLabv3 is the segmentation network. "Time / step" is measured as the ratio with respect to the default setting. * denotes the default setting.

we present a simple strategy to trade off the training efficiency and overall performance: gradient matching can be done only on the part of segmentation network parameters. Table 2 presents an ablation study about matching blocks on Cat-16, where StyleGAN2 is the generator and DeepLabv3 is the segmentation network. We consider network blocks in DeepLabv3 with ResNet-101 as backbone from bottom to top: Res1, Res2, Res3, Res4 and Res5 that are residual blocks in the backbone, and Head that is the segmentation head. Results show that skipping the bottom blocks for gradient matching improves the training speed while the segmentation performance is affected little. We conjecture the reasons can be two folds: (i) the backbone parameters are pre-trained such that little needs to be changed during training; (ii) the gradients accumulate more randomness as they backpropagate to the bottom blocks of deep neural networks. Therefore, in practice, we only match the gradients of Head in DeepLabv3 by default. Additional results are available in the appendix (see Section C.3).

## 5 CONCLUSION

We propose a gradient-matching-based method to learn annotator, which can label generated images with part segmentation by decoding the generator features into segmentation masks. Unlike existing methods that require labelling generated images for training an annotator, our method allows a broader range of labelled data, including realistic images, synthetic images, and even rendered images from 3D CAD models. On the benchmark of semi-supervised part segmentation, our method significantly outperforms other semi-supervised segmentation methods under the circumstances of extremely limited labelled data but becomes less competitive under large-scale settings, which requires future research. The effectiveness of our method is validated on a variety of single-class datasets, including well-aligned images, *e.g.* CelebA and Face-34, as well as images in the wild part of which contain cluttered backgrounds, *e.g.* Pascal-Horse, Pascal-Aeroplane, Car-20 and Cat-16. In terms of more complex scenes, discussion and investigation are further needed. With the rapid progress of generative modelling, it is promising to use more powerful generative models and explore more computer vision problems under different challenging settings in the future.

ACKNOWLEDGEMENTS

This work was supported by the National Key R&D Program of China under Grant 2018AAA0102801, National Natural Science Foundation of China under Grant 61620106005, and Beijing Municipal Science and Technology Commission grant Z201100005820005. Hakan Bilen is supported by the EPSRC programme grant Visual AI EP/T028572/1.

REPRODUCIBILITY STATEMENT

The appendix provides all the details about the datasets, network structure, and training curriculum. Code to reproduce our main results is publicly available at https://github.com/yangyu12/lagm.

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

# A  DATASET DETAILS

**CelebA**  Following Li et al. (2021), we only consider 8 part classes: background, ear, eye, eyebrow, skin, hair, mouth, and nose. Table A.1(Left) presents the protocol for merging the 19 part classes into 8 classes. CelebAMask-HQ contains 30,000 annotated images in total. We split this dataset into the unlabeled set, training set, validation set and test set as in Table A.1(Right).

| 8 classes | 19 classes | | Split | Image id |
|---|---|---|---|---|
| background | background, hat, ear_r, neck_l, neck, cloth | | Unlabeled set | 1∼28,000 |
| ear | l_ear, r_ear | | Training set (2) | 28,001∼28,002 |
| eye | l_eye, r_eye | | Training set (10) | 28,001∼28,010 |
| eyebrow | eyebrow | | Training set (30) | 28,001∼28,030 |
| skin | rest | | Training set (150) | 28,001∼28,150 |
| hair | hair | | Training set (1500) | 28,001∼29,500 |
| mouth | mouth, u_lip, r_lip | | Validation set | 27,501∼28,000 |
| nose | nose | | Test set | 29,501∼30,000 |

Table A.1: *Left*: Protocol for merging part classes on CelebA. *Right*: Dataset split on CelebA.

**Pascal-Horse & Pascal-Aeroplane**  Pascal Part (Chen et al., 2014) provides part segmentation annotations of 20 object classes for images in Pascal VOC 2010. Following other work on part segmentation (Tsogkas et al., 2015; Zhao et al., 2019; Tritrong et al., 2021), we merge the fine-grained part classes into 6 classes for horse and aeroplane. The merging protocol is presented in Table A.2. We crop the image patches that contain the object of interest according to bounding box annotations. Concretely, we discard bounding boxes with IoU with other boxes smaller than 0.05 to ensure a single object appears in a single image patch. The patches that have any side less a certain number of pixels (32 for horse; 50 for aeroplane) are also abandoned. The crop regions are extended from bounding boxes to square boxes, and the regions outside images are padded with zeros. These processed patches are finally resized to $256 \times 256$ resolution to serve the training and evaluation process. For horse and aeroplane, We further split the patches in official VOC 2010 `train` split in *training* and *validation* set, and take the patches in official VOC 2010 `val` split as *test* set. This procedure finally provides 180 images as the training set (*i.e.* labelled set), 33 images as a validation set, 223 images as the test set in Pascal-Horse, 180 images as labelled set, 78 images as the validation set and 266 images as the test set in Pascal-Aeroplane. As unlabeled images, we centre-crop the 200,000 images in the large-scale LSUN (Yu et al., 2015) archive and resize them to $256 \times 256$ resolution for both horse and aeroplane.

| Pascal-Horse | | Pascal-Aeroplane | |
|---|---|---|---|
| 6 classes | fine-grained classes | 6 classes | fine-grained classes |
| background | background | background | background |
| head | head, leye, reye, lear, rear, muzzle | body | body |
| torso | torso | stern | stern, tail |
| legs | lfho, rfho, lbho, rbho, lfuleg, lflleg, rfuleg | wing | lwing, rwing |
| neck | neck | engine | engine_{d} |
| tail | tail | wheel | wheel_{d} |

Table A.2: Protocol for merging part classes on Pascal Part.

**CGPart**  As a part segmentation dataset composed of 3D CAD models from 5 vehicle categories with 3D part manual annotations, CGPart (Liu et al., 2021) manage to render a large amount of image dataset with part mask labels by adopting blender, a well-known graphical software. In this paper, we select category *car* as the training and evaluation dataset and make moderate modifications upon the rendering pipeline. Parts are merged into 15 classes, where excessively fine-grained parts that seldom emerge are combined to avoid sample imbalance. Part merging protocol is presented in Table A.3. Viewpoints are sampled around the object with azimuth ranging in full 360°. Elevation angle is varied from 0° to 80°, yet higher sampling probability is assigned to interval [0°, 30°]. The distance

from the object to the viewpoint is carefully set to be located at the centre and fully in frame. Random colour is painted upon the object, and random texture sampled from COCO dataset (Lin et al., 2014), and Pixar-One-Twenty-Eight (Sisson, 2018) is rendered on the ground and the surroundings. $4,000$ images of $512 \times 384$ are rendered as a full train dataset. When training, only $1,000$ of them are randomly sampled as official trainset. The original `val` split of CGPart is adopted as *test* dataset, consisting of $40$ images. Test masks are merged under the same merging protocol in the inference and evaluation phases.

| CGPart-Car | |
|---|---|
| 15 classes | fine-grained classes |
| background | background |
| back_bumper | back_bumper |
| car_body | left_frame, right_frame |
| door | back_left_door, back_right_door, front_left_door, front_right_door |
| front_bumper | front_bumper |
| head_light | left_head_light, right_head_light |
| hood | hood |
| licence_plate | back_license_plate, front_license_plate |
| mirror | left_mirror, right_mirror |
| roof | roof |
| tail_light | left_tail_light, right_tail_light |
| trunck | trunk |
| wheel | back_left_wheel, back_right_wheel, front_left_wheel, front_right_wheel |
| window | back_left_window, back_right_window, front_left_window, front_right_window, left_quarter_window, right_quarter_window |
| windshield | back_windshield, front_windshield |

Table A.3: Protocol for merging part classes on CGPart.

# B IMPLEMENTATION DETAILS

**Gradient matching loss** Following DC (Zhao et al., 2021), the distance between gradients is measured with cosine similarity. The computation of gradient matching is quite similar as in DC. Here, we re-state this procedure for clarity. In particular, considering a multi-layer neural network $S_\theta$ that is parameterized with $\theta$, the gradient matching loss is computed as an average over layerwise losses as $D(\nabla_\theta \mathcal{L}_l, \nabla_\theta \mathcal{L}_g) = \frac{1}{L} \sum_{i=1}^{L} d(\nabla_{\theta^{(i)}} \mathcal{L}_l, \nabla_{\theta^{(i)}} \mathcal{L}_g)$, where $i$ denotes the layer index, $L$ denotes the number of layers for gradient matching and

$$d(\boldsymbol{A}, \boldsymbol{B}) = \sum_{j=1}^{N} \left( 1 - \frac{\boldsymbol{A}_j \cdot \boldsymbol{B}_j}{\|\boldsymbol{A}_j\| \cdot \|\boldsymbol{B}_j\|} \right) \tag{12}$$

where $\boldsymbol{A}_j$ and $\boldsymbol{B}_j$ are flattened gradient vectors for each neural node $j$. The minor difference of our practice compared DC is that we average the layerwise gradient distances whereas DC sum them. It is easier to handle numerical issues using our practice in our problem. In terms of gradient matching for normalization layers such as Batch Normalization (BN), we follow the practice of DC to ignore the gradient matching of learnable parameters in BN. The BN layer is set as *train* mode during gradient matching. We find it works well.

**Pre-trained GAN** We use StyleGAN family (Karras et al., 2019; 2020b;a) as our pre-trained GANs. We either train GANs by ourselves for each dataset or use publicly available models. The training configurations or links of the pre-trained GANs are listed in Table B.4. For GANs trained by ourselves, we use the checkpoints with the lowest FID in the historical ones.

**Annotator structure** An annotator is a neural network that takes the generator features and output segmentation masks as input. DatasetGAN (Zhang et al., 2021) and RepurposeGAN (Tritrong et al., 2021) first upsample feature maps to full resolution (resolution of output image) and then concatenate

| Dataset | Version | Output resolution | Training configurations or links |
|---|---|---|---|
| CelebA | StyleGAN2-ADA | $256 \times 256$ | `--cfg=paper256 --mirror=True` |
| Pascal-Horse | StyleGAN2 | $256 \times 256$ | stylegan2-horse-config-f.pkl |
| Pascal-Aeroplane | StyleGAN2 | $256 \times 256$ | `--cfg=stylegan2 --aug=noaug` |
| Cat-16 | StyleGAN | $256 \times 256$ | available in DatasetGAN repo |
| Cat-16 | StyleGAN2 | $256 \times 256$ | lsuncat200k-paper256-ada.pkl |
| Face-34 | StyleGAN | $512 \times 512$ | available in DatasetGAN repo |
| Face-34 | StyleGAN2-ADA | $512 \times 512$ | `--cfg=paper512 --mirror=True` |
| Car-20 | StyleGAN | $512 \times 512$ | available in DatasetGAN repo |
| Car-20 | StyleGAN2 | $512 \times 512$ | stylegan2-car-config-f.pkl |

Table B.4: Pretrained GANs. We use the source codes provided by Karras et al. (2020a) to train Style-GAN models: `https://github.com/NVlabs/stylegan2-ada-pytorch`. For CelebA and Face-34 the StyleGAN-ADA is trained on CelebAHQ-Mask 28k images at $256 \times 256$ and $512 \times 512$ resolution, respectively. For Pascal-Aeroplane, the StyleGAN2 is trained on 200k images from LSUN airplane.

these feature maps along the channel. This procedure results in highly high-dimensional feature vectors for every pixel. These features are fed into either multiple-layer perceptrons (MLPs) as in DatasetGAN (Zhang et al., 2021) or convolutional neural networks (CNNs) as in (Tritrong et al., 2021). Nonetheless, this practice consumes a lot of GPU memory and run painfully slow even on high-end modern GPUs. That is also why DatasetGAN cannot consume all image feature vectors in a batch during training.

Therefore, to make training annotators more efficient, as commonly done in segmentation or detection network architectures, we fuse the multi-scale feature maps with a feature pyramid network (FPN) (Lin et al., 2017) structure and decode the fused features into segmentation masks with consecutive convolutional layers. This FPN structure saves a lot of GPU memories and computation, which allows us to consume multiple full images in one mini-batch during training. An illustration of our annotator structure is presented in Figure B.1. For generator features that are fed into annotator, we use outputs of all convolutional layers in StyleGAN as in DatasetGAN (Zhang et al., 2021) and use outputs of all synthesis blocks, each of which typically contains two consecutive convolutional layers, in StyleGAN2.

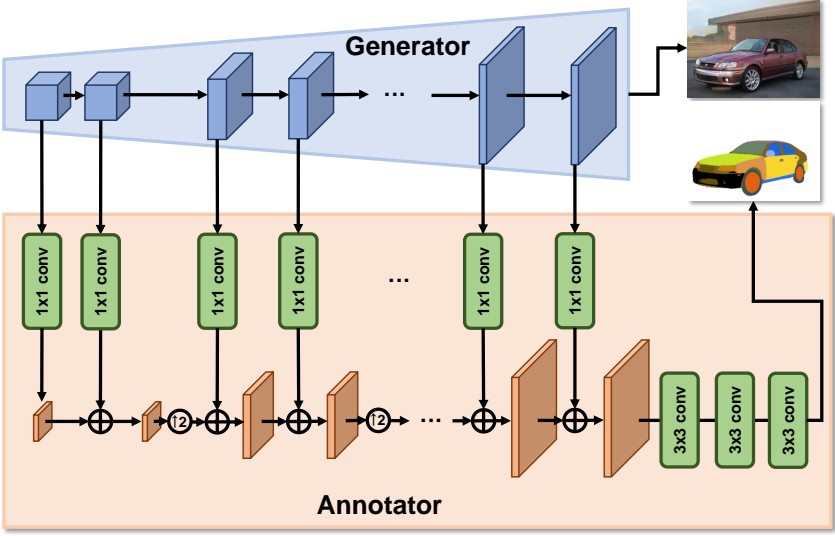

Figure B.1: Annotator architecture.

**Segmentation network structures** In our experiments, we employ two prevalent segmentation network structures: U-Net (Ronneberger et al., 2015) and DeepLabv3 (Chen et al., 2017) with ResNet-101 pretrained on ImageNet as backbone. For U-Net, we reference the implementation from `https://github.com/milesial/Pytorch-UNet` and train it from scratch on target datasets. For DeepLabv3, we use the the built-in implementation in PyTorch library[4]. For all the experiments on CelebA, Pascal-Horse, and Pascal-Aeroplane, we report the results (Figure 2) using DeepLabv3 as segmentation networks.

**Training details of our method** We use pixelwise cross-entropy loss as $f(\cdot, \cdot)$ for training segmentation network and computing gradients. The annotator and the segmentation network are optimized with an SGD optimizer with learning rate 0.001 and momentum 0.9. By default, we jointly train an annotator and a segmentation network with $K = 1$ and batch size 2 for $150,000$ steps.

**Training details of competing semi-supervised methods** For MT, GCT and AdvSSL, we train the networks for $10,000$ iterations with batch size 16 for labelled data and 8 for unlabeled data. The learning rate is decayed with a power of 0.9 every iteration. During the training process, the input images are randomly resized to a scale in $[160, 640]$ and cropped to $256 \times 256$. We use the default settings for the rest of the hyper-parameters as in PixelSSL.

## C FURTHER ANALYSIS

### C.1 COMPARISON TO OTHER BASELINES

**Inversion method** As discussed in Section 1, the "inversion method" is a straightforward way to utilize existing labelled images to train annotators in the supervised learning manner. We evaluate the performance of the inversion method and compare it to our method on Cat-16, Face-34, and Car-20 using StyleGAN2 as the generator. In particular, inversion baseline first projects images into GAN latent space using the projection method provided by Karras et al. (2020b). Then the latent style codes are forwarded to the generator to acquire generator features. Finally, it trains the annotator using the generator features and ground truth masks.

Table C.5 shows the quantitative results and Figure C.2 presents examples of reconstruction quality. Despite its accurate image reconstruction, the inversion method fails to combat our method and produces lower-quality automatic labels than ours (see Figure C.3). We believe it is non-trivial to inverse the GAN generation process to acquire features for specific images. Even though the reconstruction of images is satisfying, one can still recover inaccurate generator features, leading to degraded annotator learning.

| Methods | $G$ | $S$ | Cat-16 | Face-34 | Car-20 |
|---------|-----|-----|--------|---------|--------|
| Ours | StyleGAN2 | DeepLab | $33.56 \pm 0.17$ | $55.10 \pm 0.39$ | $61.21 \pm 2.07$ |
| | StyleGAN2 | U-Net | $31.90 \pm 0.75$ | $53.58 \pm 0.45$ | $58.30 \pm 2.64$ |
| Inversion method | StyleGAN2 | DeepLab | $15.14 \pm 0.27$ | $47.87 \pm 0.86$ | $52.18 \pm 2.31$ |
| | StyleGAN2 | U-Net | $12.78 \pm 0.42$ | $48.57 \pm 0.38$ | $52.41 \pm 1.69$ |

Table C.5: Comparison of our methods to inversion method.

**Pseudo-labeling method** Another way to train annotators using existing labeled images in supervised learning manner is pseudo-labeling method, which uses a trained segmentation model to predict pseudo labels for the generated images. In particular, this method involves three stages. (i) A segmentation model is trained with the labeled dataset. (ii) The trained segmentation model is used to produce pseudo labels for every random image generated from generator. (iii) The generator features and pseudo labels are used to train annotators in the supervised learning manner.

Another way to train annotators using existing labelled images in a supervised learning manner is the pseudo-labelling method, which uses a trained segmentation model to predict pseudo labels for the generated images. In particular, this method involves three stages. (i) A segmentation model is

---

[4]`https://pytorch.org/hub/pytorch_vision_deeplabv3_resnet101/`

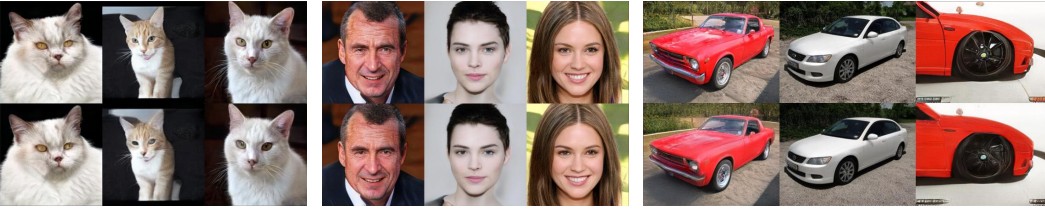

Figure C.2: Examples of reconstruction quality. The first row shows target images and the second row shows the reconstructed images.

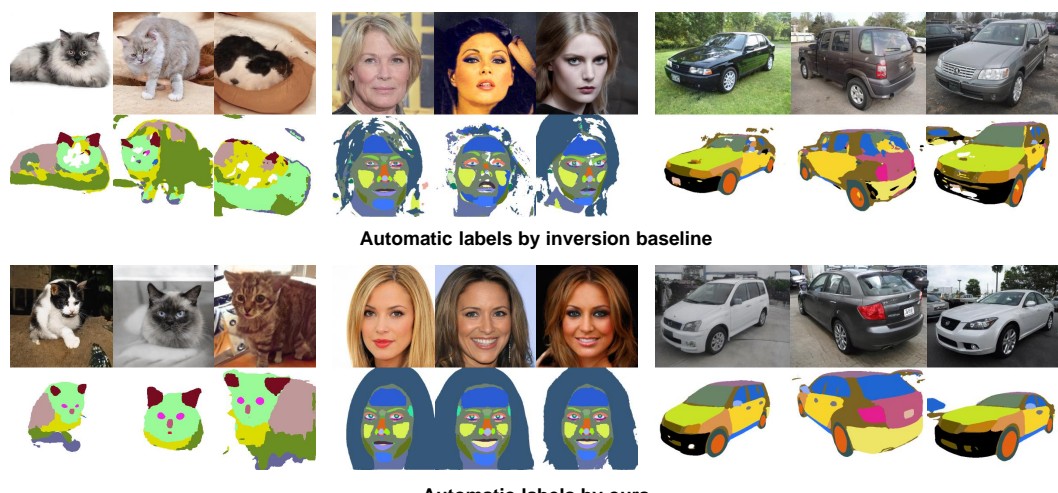

Figure C.3: Comparison of the automatic labels produced by inversion method v.s. ours.

trained with the labelled dataset. (ii) The trained segmentation model produces pseudo labels for every random image generated from the generator. (iii) The generator features and pseudo labels are used to train annotators in supervised learning.

| Methods | $G$ | $S$ | Cat-16 | Face-34 | Car-20 |
|---|---|---|---|---|---|
| Ours | StyleGAN2 | DeepLab | $33.56 \pm 0.17$ | $55.10 \pm 0.39$ | $61.21 \pm 2.07$ |
| | StyleGAN2 | U-Net | $31.90 \pm 0.75$ | $53.58 \pm 0.45$ | $58.30 \pm 2.64$ |
| Pseudo-labeling method | StyleGAN2 | DeepLab | $20.21 \pm 0.65$ | $42.15 \pm 0.68$ | $12.37 \pm 0.91$ |
| | StyleGAN2 | U-Net | $16.72 \pm 0.79$ | $43.26 \pm 1.20$ | $13.48 \pm 0.48$ |

Table C.6: Comparison of our methods to pseudo-labeling method.

**Our method with MAML** One straightforward solution to Equation 6 is to use a MAML-like algorithm, which unrolls the inner loop with multi-step stochastic gradient descent, reserves the computation graph, computes the meta loss, and finally backpropagates the gradients for updating the annotator. We refer to this method as "Ours-$K$-step-MAML" and summarize this algorithm as Algorithm 2. Table C.7 shows the experimental comparison of gradient matching versus MAML, where we use $\eta_i = 0.1$ in MAML for all experiments. The following conclusions can be drawn from these results. First, gradient matching generally leads to higher performance and obtains more consistent results across different datasets and different segmentation network architectures than MAML. Second, MAML probably requires multiple inner-loop steps (more than two) to match the performance of gradient matching, given that it fails to do so with two inner-loop steps. It is very inefficient and requires enormous computation resources. Furthermore, in practice, we observe that MAML appears unstable and sensitive to hyperparameters (*e.g.* inner-loop learning rate) across different datasets and different network architectures, whereas gradient matching requires neither dataset-specific nor network-architecture-specific hyperparameters.

**Algorithm 2:** Learning to annotate with $K$-step MAML.

**Inputs :**

    $G$               trained generator

    $\mathcal{D}_l$             set of labeled examples

    $\omega, A_\omega, \eta_\omega$    initial annotator parameters, annotator, and learning rate for annotator

    $\theta, S_\theta, \eta_\theta$     initial segmentor parameters, segmentor, and learning rate for segmentor

    $T$ and $B$     total number of optimization steps and batch size

    $\eta_i$ and $K$     inner-loop learning rate and inner-loop steps

1 **for** $t \leftarrow 1$ **to** $T$ **do**

    `// update annotator` $A_\omega$

2     $\theta' \leftarrow \theta$                 `// copy segmentor parameters for simulated SGD`

3     $\mathcal{B}_g \leftarrow \{(\mathbf{x}_j, A_\omega(\mathbf{h}_j))\}_{j=1}^{B}$         `// sample a batch of synthetic data`

4     **for** $k \leftarrow 1$ **to** $K$ **do**

5         $\nabla_{\theta'}\mathcal{L}_g \leftarrow \frac{1}{B}\sum_{j=1}^{B}\nabla_{\theta'}f(S_{\theta'}(\mathbf{x}_j), A_\omega(\mathbf{h}_j))$    `// compute gradients on` $\mathcal{B}_g$

6         $\theta' \leftarrow \theta' - \eta_i\nabla_{\theta'}\mathcal{L}_g$           `// update segmentor parameters`

7     **end**

8     $\mathcal{B}_l \leftarrow \{(\mathbf{x}_i, \mathbf{y}_i) \sim \mathcal{D}_l\}_{i=1}^{B}$      `// sample a batch of labeled examples`

9     $\mathcal{L}_l \leftarrow \frac{1}{B}\sum_{i=1}^{B}f(S_\theta(\mathbf{x}_i), \mathbf{y}_i)$         `// compute meta loss`

10     $\omega \leftarrow \omega - \eta_\omega\nabla_\omega\mathcal{L}_l$           `// update annotator parameters`

    `// update segmentor` $S_\theta$

11     $\mathcal{B}_g \leftarrow \{(\mathbf{x}_j, A_\omega(\mathbf{h}_j))\}_{j=1}^{B}$         `// sample a batch of synthetic data`

12     $\mathcal{L}_g \leftarrow \frac{1}{B}\sum_{j=1}^{B}f(S_\theta(\mathbf{x}_j), A_\omega(\mathbf{h}_j))$     `// compute loss on synthetic data`

13     $\theta \leftarrow \theta - \eta_\theta\nabla_\theta\mathcal{L}_g$           `// update segmentor parameters`

14 **end**

**Output :** annotator $A_\omega$ and segmentor $S_\theta$

| Methods | $G$ | $S$ | Cat-16 # annotations: 30 # classes: 16 | Face-34 # annotations: 16 # classes: 16 | Car-20 # annotations: 16 # classes: 20 |
|---|---|---|---|---|---|
| Ours-GM | StyleGAN | DeepLab | $33.89 \pm 0.43$ | $52.58 \pm 0.61$ | $63.55 \pm 2.25$ |
| | StyleGAN | U-Net | $32.64 \pm 0.74$ | $53.69 \pm 0.54$ | $60.45 \pm 2.42$ |
| | StyleGAN2 | DeepLab | $33.56 \pm 0.17$ | $55.10 \pm 0.39$ | $61.21 \pm 2.07$ |
| | StyleGAN2 | U-Net | $31.90 \pm 0.75$ | $53.58 \pm 0.45$ | $58.30 \pm 2.64$ |
| Ours-1-step-MAML | StyleGAN | DeepLab | $32.49 \pm 0.43$ | $35.24 \pm 0.20$ | $54.74 \pm 2.67$ |
| | StyleGAN | U-Net | $16.70 \pm 0.43$ | $23.21 \pm 0.07$ | $29.26 \pm 0.72$ |
| | StyleGAN2 | DeepLab | $30.30 \pm 0.53$ | $33.82 \pm 0.24$ | $54.42 \pm 3.11$ |
| | StyleGAN2 | U-Net | $22.78 \pm 0.65$ | $32.64 \pm 0.18$ | $26.59 \pm 0.60$ |
| Ours-2-step-MAML | StyleGAN | DeepLab | $32.98 \pm 0.50$ | $39.73 \pm 0.13$ | $58.22 \pm 2.17$ |
| | StyleGAN2 | DeepLab | $29.66 \pm 0.69$ | $39.04 \pm 0.23$ | $56.47 \pm 2.32$ |

Table C.7: Comparisons of our methods using gradient matching (GM) versus MAML on Car-20, Cat-16, Face-34 with different network architectures. The performance is evaluated with mean intersection over union (mIoU(%)) across all part classes plus a background class.

## C.2 THE IMPACT OF SEGMENTATION NETWORK STATES

We study the impact of segmentation network states, *i.e.* segmentation parameters, in the alternate learning algorithm. In particular, we compare the training the annotators' training convergence and the performance when different series of segmentation network states are employed for gradient matching. Figure C.4 & Figure C.5 shows the evolution of annotator performance during training process, where "baseline" denotes the default setting and the following settings are further compared. (i) "random re-init": the segmentation network parameters are randomly re-initialized rather than learned from generated images. (ii) "pre-trained": the segmentation network is pre-trained to acquire an acceptable performance ($\sim 40.7\%$ mIoU evaluated on the test set). (iii) "scratch": the segmentation network is a DeepLabv3 network with a randomly initial backbone rather than pre-trained on ImageNet.

(iv) "random init & fixed" and "pre-trained & fixed": the segmentation network parameters, either randomly initialized or pre-trained, are fixed, respectively.

First, the "fixed" segmentation network results in a gradient matching problem on a single segmentation network state, leading to an annotator with inferior performance than the default setting. This result suggests that matching gradients for various segmentation network states is vital to learning annotators. Second, even though requiring an annotator to produce segmentation-network-agnostic labels is crucial, a random re-initialization strategy does not work well (see results of "random re-init" v.s. "baseline"). Moreover, the results of "pre-trained & fixed" are better than "random init & fixed", which suggest that a well-performed segmentation network provides more informative gradients such that by gradient matching, the annotator can be more effectively learned. Third, matching gradients of pre-trained segmentation networks accelerates the learning of annotators (see results of "baseline" v.s. "pre-trained" and "baseline" v.s. "scratch"), suggesting a well-performed segmentation network provides more informative gradients.

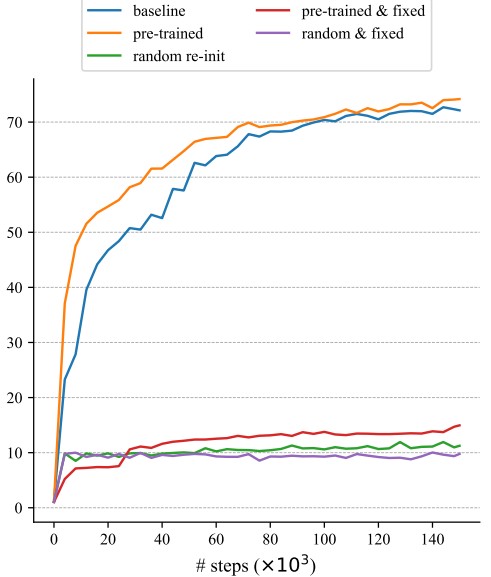 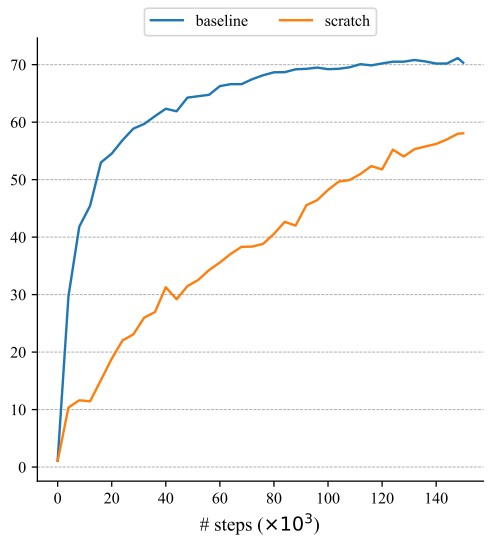

Figure C.4: Comparison of the training convergence under different settings of sampling segmentation network states. Annotator performance (mIoU (%)) is evaluated on training set of Car-20. Segmentation network is U-Net.

Figure C.5: Comparison of the training convergence of segmentation network with pretrained backbone ("baseline") versus segmentation network with randomly initialized backbone ("scratch"). Annotator performance (mIoU (%)) is evaluated on the training set of Car-20. The segmentation network is DeepLabv3.

### C.3 TRADE OFF TRAINING EFFICIENCY WITH PERFORMANCE

**Gradient matching on partial network parameters** We present additional results on in Table C.8. U-Net takes as input an image and first produce a feature map at multiple downsampled scales with consecutive convolutional and pooling layers. These feature maps are then processed by consecutive convolutional and upsampling layers to be mapped back to the original input scale. Notably, skip connection is employed to connect the multi-scale feature maps along the downsampling and upsampling paths. We consider the following particular group of layers in U-Net: (i) `Up` v.s. `Down`, (ii) `Input` v.s. `Output`, and (iii) `U1` v.s. `U2`. `Up` and `Down` denote the convolutional layers along the upsampling and downsampling path, respectively. `Input` and `Output` denote the input convolutional layer and the output convolutional layer, respectively. `U1` and `U2` denote the convolutional layers that process feature maps at $1/1$ and $1/2$ scales, respectively. The results show that matching gradients of layers near the output end is more effective than doing so at layers near the input end.

| Blocks | * All | Up | Down | Input | Output | U1 | U2 |
|---|---|---|---|---|---|---|---|
| Time / step | 1× | 0.65× | 0.73× | 0.73× | 0.34× | 0.71× | 0.63× |
| mIoU (%) | 31.90 ±0.75 | 30.12 ±0.43 | 14.42 ±0.23 | 12.40 ±0.40 | 25.41 ±0.77 | 30.18 ±0.79 | 31.22 ±0.48 |

Table C.8: **Ablation study w.r.t. blocks for matching gradients** on U-Net. Evaluation is performed on Cat-16, where StyleGAN2 is the generator. "Time / step" is measured as the ratio to the case of default setting. * denotes the default setting.

**Input resolution to segmentation network** Another way to improve the training efficiency is to reduce the input resolution to the segmentation network. Figure C.6 and Table C.9 show that reducing the resolution improves the training speed but compromises the annotator performance and downstream segmentation network performance. Therefore, in practice, the input resolution to the segmentation network can be tuned to trade off the training efficiency versus effectiveness.

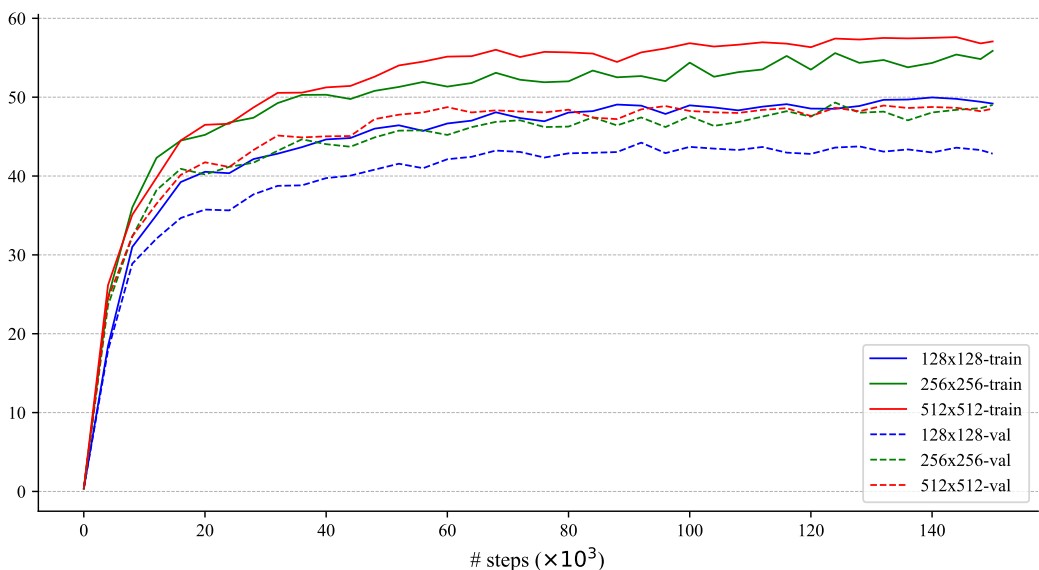

Figure C.6: The evolution of annotator performance(mIoU (%)) evaluated on training and validation set during the training process on Face-34.

| Resolution | * 512 × 512 | 256 × 256 | 128 × 128 |
|---|---|---|---|
| Time / step | 1× | 0.45× | 0.37× |
| mIoU (%) | 51.93 ± 0.28 | 50.71 ± 0.39 | 45.53 ± 0.31 |

Table C.9: Comparisons of the training efficiency (time / step) and downstream segmentation (U-Net) performance (mIoU (%)) w.r.t. different input resolution to segmentation network on Face-34. "Time / step" is measured as ratio with respect to the default setting. * denotes the default setting.

## C.4 THE IMPACT OF GENERATOR

As our segmentation network is trained only on the synthetic images and labels (Algorithm 1 Line 9∼11), the segmentation performance is inevitably affected by the quality of pre-trained GANs. Table C.10 confirms this point and shows that premature GANs – ones with high FID score – generally leads to lower segmentation performance.

|  | CelebA | | | | Pascal-Aeroplane | | | |
|---|---|---|---|---|---|---|---|---|
| FID $\downarrow$ | 10.21 | 6.80 | 5.12 | 4.29 | 11.74 | 7.78 | 6.25 | 5.04 |
| FG-mIoU (%) $\uparrow$ | 73.92 | 76.61 | 77.48 | 78.07 | 21.69 | 36.01 | 38.81 | 41.21 |

Table C.10: The segmentation performance with respect to generators of different quality (indicated by FID) on CelebA and Pascal-Aeroplane.

## C.5 DISCUSSION ON LARGE-SCALE SETTING

As shown in Fig. 2, the performance of our method begins to saturate and even be beaten by other semi-supervised segmentation methods when the number of labels grows large. To investigate the performance of our method under a large-scale setting, we further run experiments with more unlabeled and labelled data and present the results in Table C.11. It shows that the performance of our method even slightly drops when the number of unlabeled data or the number of labels increases and is significantly beaten by supervised learning. We hypothesize the reason to be that our method is seriously limited by the performance of GANs, which still struggle to fit large-scale datasets. In contrast, supervised learning becomes stronger under a large-scale setting since more labels are available. How to address this issue requires further research in the future.

| Method | unlabeled data | # labels | FG-mIoU (%) |
|---|---|---|---|
| Supervised learning | – | 29,000 | 84.32 |
| Ours | CelebA-train (29,000) | 1,500 | 77.90 |
| Ours | CelebA-train (29,000) | 29,000 | 76.03 |
| Ours | CelebA-train + FFHQ (99,000) | 29,000 | 75.00 |

Table C.11: Experiments on human face part segmentation under large-scale setting.

## C.6 COMPARISON TO DATASETGAN IN OTHER ASPECTS

**Real images v.s. synthetic images as labeled data** As our method removes the necessity of labelling synthetic images, we investigate if replacing the labelled synthetic images with real ones could improve the segmentation performance. Table C.12 shows an evaluation on CelebA-test at $512\times512$ resolution across 8 face classes. DatasetGAN (Zhang et al., 2021), which uses 16 annotated synthetic images for training, achieves 70.01 mIoU performance. Our method achieves a bit higher performance when using the exactly same synthetic images as DatasetGAN. Moreover, the performance of our method can be further improved with around 6.7% mIoU by replacing the labelled synthetic images with real ones in CelebA-train. This result suggests that the capability of using real labelled data exhibits the superiority of our method over DatasetGAN.

|  | Labeled data | CelebA-test@$512\times512$ |
|---|---|---|
| DatasetGAN (Zhang et al., 2021)[†] | Synthetic | 70.01 |
| Ours | Synthetic | 72.55 |
|  | Real | 79.25 |

Table C.12: Comparison of the performance when using synthetic images versus real images as labeled data. †: Results taken from Zhang et al. (2021).

**Our method with more labeled data on Car-20** Considering our method does not match the performance of DatasetGAN on Car-20, we further investigate if this gap can be narrowed with the help of more labelled data. Table C.13 presents the performance of our method when more labelled data is used on Car-20. It shows that with an increased number of labelled data, the performance of our method gradually approaches that of DatasetGAN.

## D QUALITATIVE RESULTS

| G | S | DatasetGAN | Ours | | |
|---|---|---|---|---|---|
| | | # annotations: 16 | # annotations: 16 | # annotations: 25 | # annotations: 33 |
| StyleGAN | DeepLab | $67.53 \pm 2.58$ | $63.55 \pm 2.25$ | $64.68 \pm 2.53$ | $64.88 \pm 2.52$ |
| StyleGAN | U-Net | $66.27 \pm 2.75$ | $60.45 \pm 2.42$ | $63.09 \pm 3.49$ | $65.37 \pm 2.78$ |

Table C.13: Performance of our method with respect to different number of labeled images on Car-20 dataset.

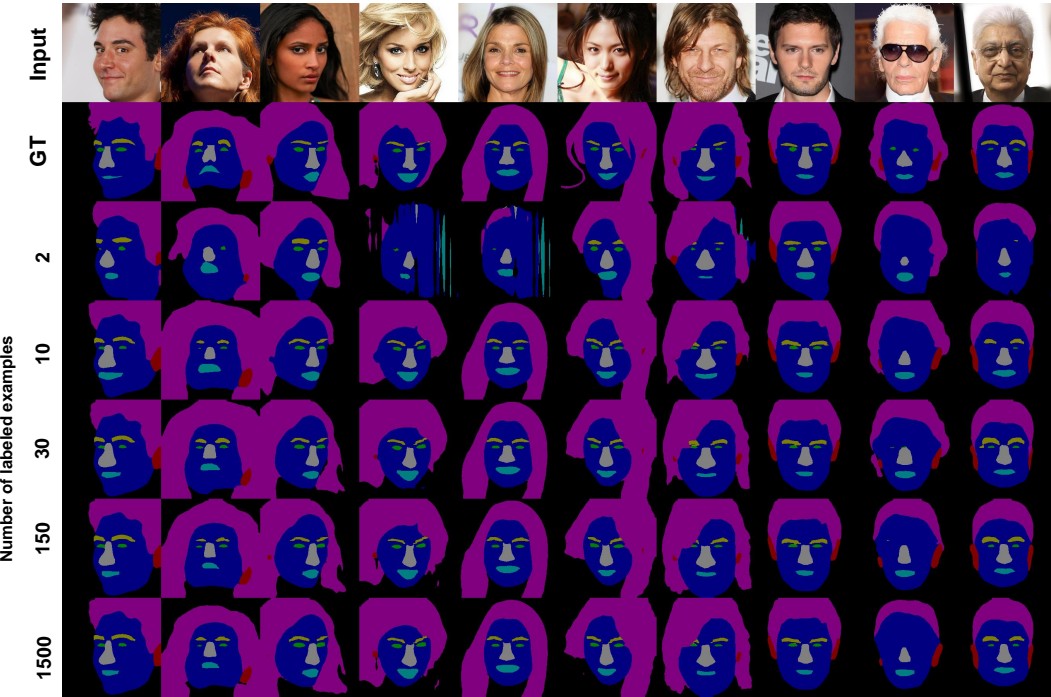

Figure D.7: Qualitative results of our method on CelebA. The segmentation results of our models trained from 2, 10, and 30, 150, 1500 labeled examples are presented.

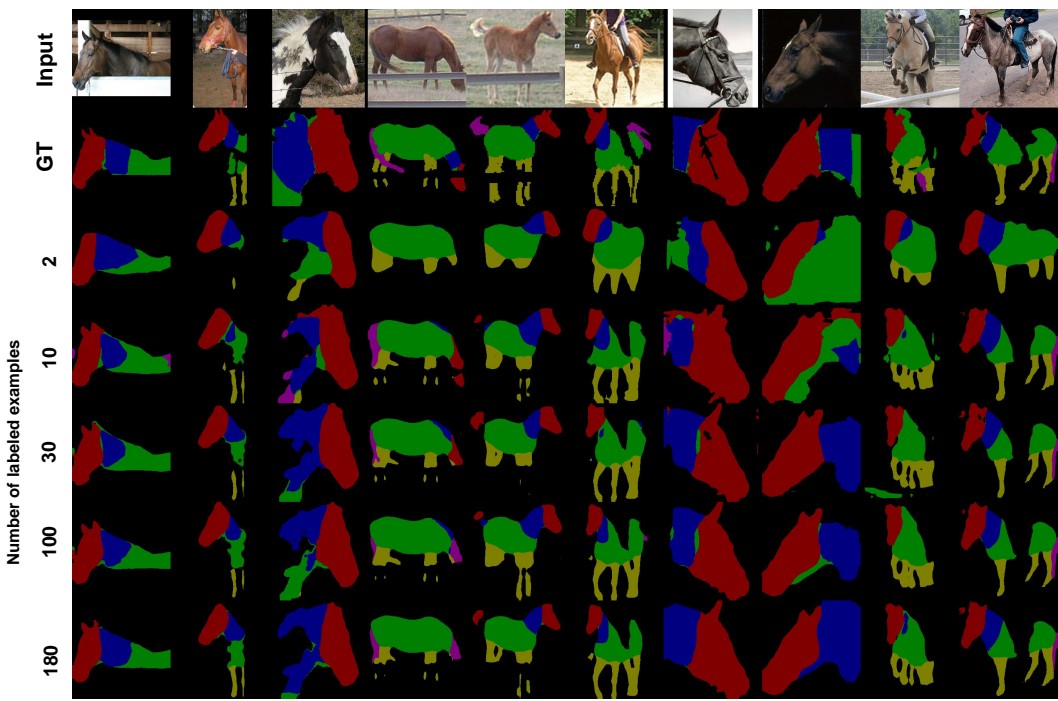

Figure D.8: Qualitative results of our method on Pascal-Horse. The segmentation results of our models trained from 2, 10, and 30, 100, 180 labeled examples are presented.

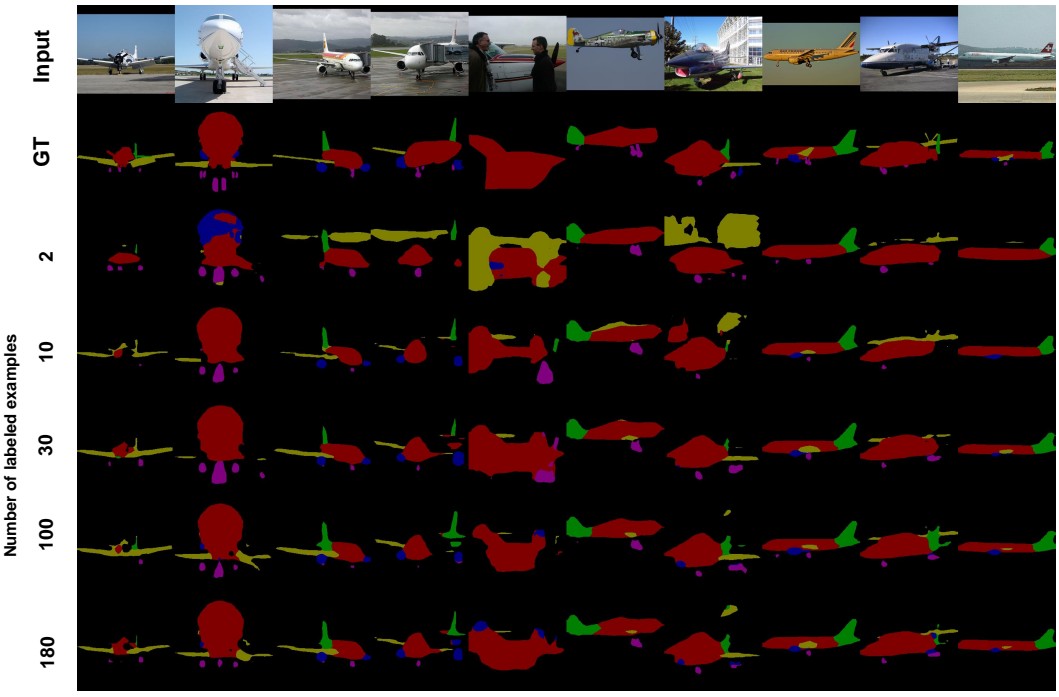

Figure D.9: Qualitative results of our method on Pascal-Aeroplane. The segmentation results of our models trained from 2, 10, and 30, 100, 180 labeled examples are presented.

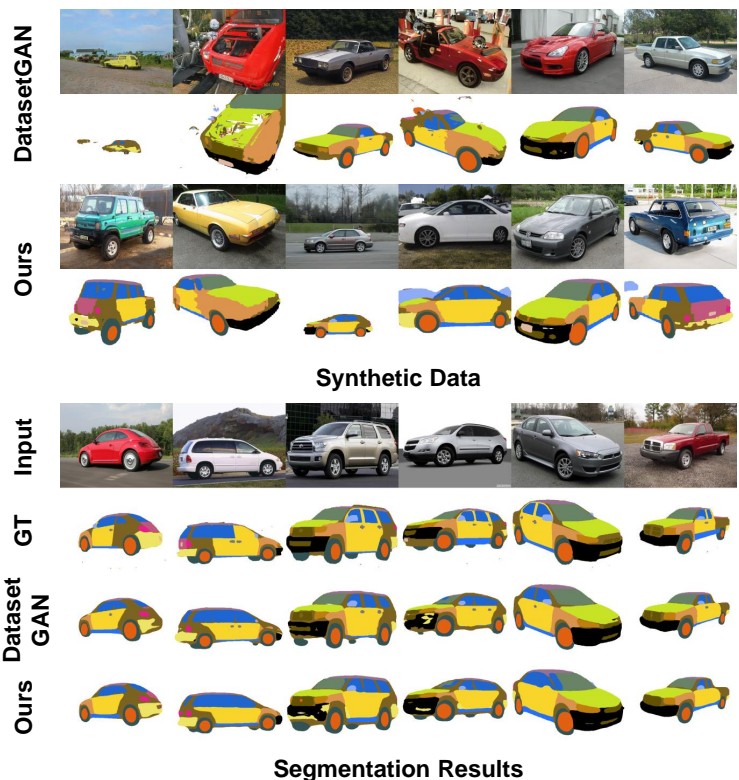

Figure D.10: Qualitative comparison of our method to DatasetGAN on Car-20

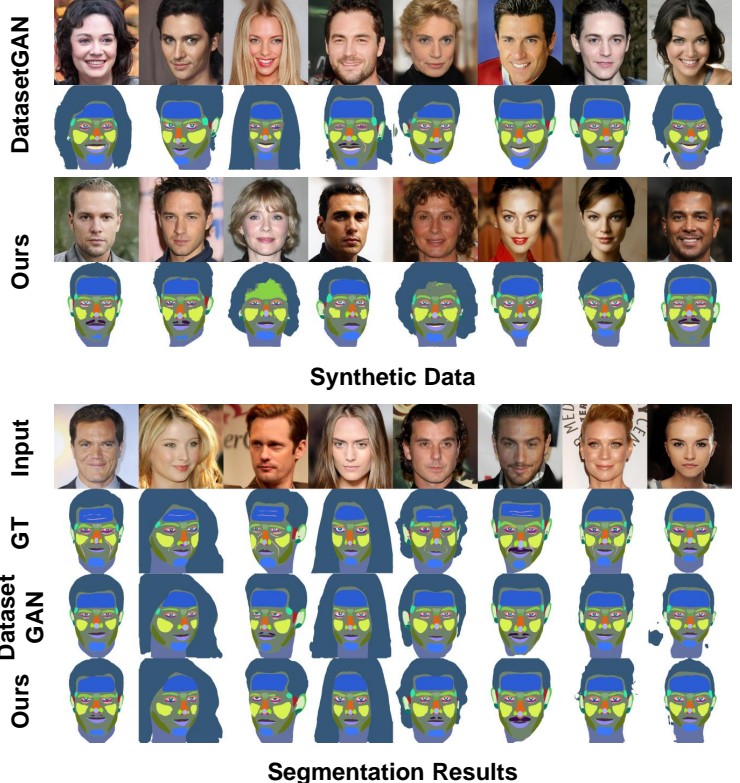

Figure D.11: Qualitative comparison of our method to DatasetGAN on Face-34.

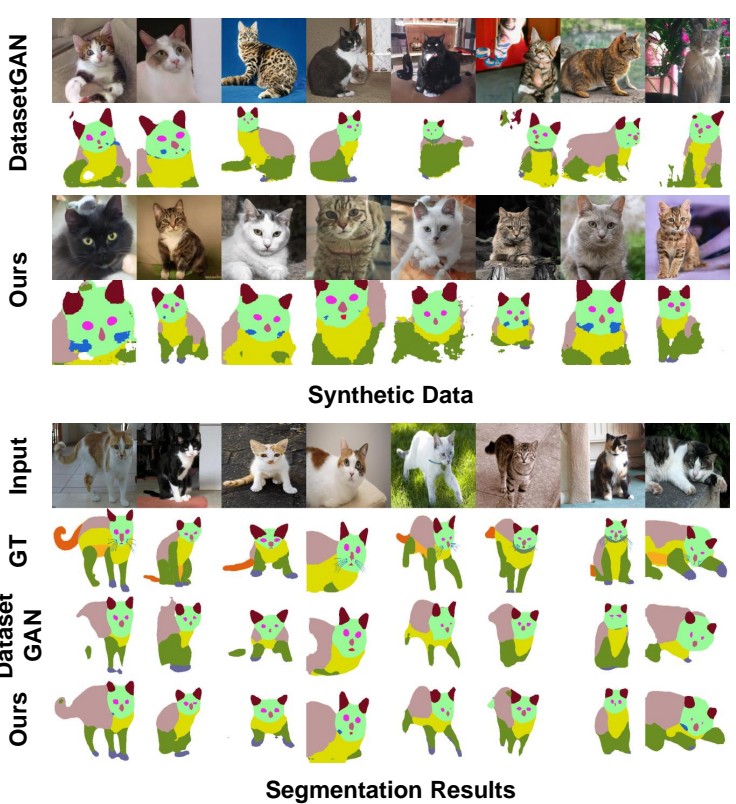

Figure D.12: Qualitative comparison of our method to DatasetGAN on Cat-16.

