# OpenReview forum: "Learning to Annotate Part Segmentation with Gradient Matching"
_ICLR.cc/2022/Conference — ICLR 2022 Poster_

### Official Review · Reviewer_muji · 2021-10-29

**Correctness:** 4
**Technical Novelty And Significance:** 2
**Empirical Novelty And Significance:** Not applicable
**Recommendation:** 6
**Confidence:** 3

**Main Review:**

Reducing the annotation cost is a very important thing. In this work, the authors provided a potential and promising way based on DatasetGAN.

Majors. (1) Contributions. In the last paragraph in Section Introduction, the authors might not very clearly summarize their contributions. I suggest they could highlight the contributions one by one. (2) Method. During the generation process in eq. (1), the authors just did the random sample or sample the z according to the labeled data? Like the examples in Fig. 1, the generated data has similar attributes with the labeled data, such as the view. (3) Experiment. (a) Datasets. The authors use three datasets in the experimental part. Maybe, it might not enough to evaluate the effectiveness. DatasetGAN, the most related work, used more than 3 datasets. (b) Dataset setting. The authors need to explain the reason why they followed (Li et al. (2021), not DatasetGAN (Zhang et al., 2021). In table 1, they just copied the numbers from DatasetGAN (Zhang et al., 2021). I suggest the authors could add a summary of all test datasets as Table 1 in (Zhang et al., 2021). (c) Table 1, the authors also need to provide more details about how to get the results of DatasetGAN, which are not the same ones in DatasetGAN.

Minors. (1) There exists some typos, such as “our method achieve same performance” -> ‘achieves’. (2) The citations of TL and SSL in Table 1 might be modified to the original work, and the authors might add some footnotes to indicate they just copied the numbers from (Zhang et al., 2021).


**Summary Of The Paper:**

In this paper, the authors mainly propose a gradient-matching-based method for part segmentation to reduce the annotation cost. Based on the DatasetGAN, the proposed model also used the Style GAN family to generate high-quality images and remove the human annotations on a handful of synthesized images. Compared with semi-supervised learning methods, their proposed method achieves higher performance on three datasets.

**Summary Of The Review:**

The contributions might need to be clearly summarised and I vote 'the marginally below the acceptance threshold', based on the limited experiment and the unclear experimental settings.

---

> ### Author Response · Authors · 2021-11-20
> **Reply to Reviewer muji**
>
> **Q1: Summary of our contributions**
>
> We have added a paragraph as a summary of our contributions.
>
> > Our contribution can be summarized as follows. (i) We formulate the learning of annotations for GAN-generated images as a learning-to-learn problem and propose an algorithm based on gradient matching to solve it. As a consequence, a broad range of labeled data including real data, synthetic data, and even out-of-domain data is applicable. (ii) We empirically show that our method significantly outperforms other semi-supervised segmentation methods under the circumstances of extremely limited labeled data.
>
>
> **Q2: During the generation process in eq. (1), the authors just did the random sample or sample the z according to the labeled data?**
>
>
> Thanks for your question. In Eq. (1), we randomly sample $z$ from a normal distribution as denoted by $z \sim~P_z$. We have changed the example images in Fig. 1 to avoid the possible wrong impression and emphasize that the synthetic batch is a random one in the caption.
>
> **Q3: The authors use three datasets in the experimental part. Maybe, it might not enough to evaluate the effectiveness. DatasetGAN, the most related work, used more than 3 datasets.**
>
> Thanks for your concerns. Actually, we evaluate our methods on six datasets and the main results in Fig.2 & Table 1 are on different datasets. In the original manuscript, three datasets including **CelebA**, **Pascal-Horse**, **Pascal-Aeroplane** in terms of real images as labeled data, and three datasets including **Car-20**, **Cat-16**, and **Face-34**（all publicly available ones in DatasetGAN）in terms of synthesized images as labeled data are not stated in the same place. We are sorry if such a description misleads you about the experimental statement. In our updated version, we have changed the style to reduce confusion.
>
> We have also enriched our experimental analysis as suggested by all the reviewers and supplemented these results to the appendix, which may hopefully address your concerns about our experiments.
>
> **Q4: The authors need to explain the reason why they followed (Li et al. (2021), not DatasetGAN (Zhang et al., 2021).**
>
> Thanks for your concerns. In fact, since our method allows both real images and synthetic images as labeled data, both settings (Li et al., 2021 & DatasetGAN) are considered in our paper. We suppose it might be our way of presentation that made you neglect this point. Therefore, the headings of paragraphs in Sec.4.2 are changed to better present the logic of our experimental settings.
>
> > Since our method relaxes the restrictions on the source of labeled data, in this section, we show the effectiveness of our method under three different circumstances: real images as labeled data, synthetic images as labeled data, and out-of-domain images as labeled data.
> >
> > **Real images as labeled data ...**
> >
> > **Synthetic images as labeled data ...**
> >
> > **Out-of-domain images as labeled data ...**
>
> **Q5: I suggest the authors could add a summary of all test datasets as Table 1 in (Zhang et al., 2021). … the authors also need to provide more details about how to get the results of DatasetGAN, which are not the same ones in DatasetGAN.**
>
> Many thanks for this valuable suggestion. We have made the suggested changes in our revised version.
>
> We report the up-to-date results in the [DatasetGAN github repository](https://github.com/nv-tlabs/datasetGAN_release) as the results of DatasetGAN-StyleGAN-DeepLab, which are higher than those in the original paper due to minor implementation differences. We have verified the reproducibility of this up-to-date performance and therefore report them. The results of DatasetGAN-StyleGAN-UNet are obtained by ourselves. We believe the results are reasonable since they are quite close to those of DatasetGAN-StyleGAN-DeepLab.
>
>
> **Q6: Typos and citation issues.**
>
> Many thanks to the reviewer for the careful check of details. Issues are fixed in our revised version.

---

> > ### Comment · Reviewer_muji · 2021-11-30
> > **Reply to Authors' Response**
> >
> > Thanks for the detailed feedback. I have carefully read the feedback and other reviews. The authors did wonderful work and have addressed my questions. I changed my rating and suggest the authors could add the new discussions and experiment in the revised version.

---

### Official Review · Reviewer_y2iP · 2021-11-02

**Correctness:** 3
**Technical Novelty And Significance:** 3
**Empirical Novelty And Significance:** 2
**Recommendation:** 6
**Confidence:** 4

**Details Of Ethics Concerns:**

None at this moment.

**Main Review:**

<Pros>


- The authors argued that the proposed method has a few benefits over the prior arts. The limitations of the prior arts (DatasetGAN and RepurposeGAN) are discussed: 1) they are only applicable to generator-specific generated data, and 2) they need manual re-labeling when the generator is changed. In contrast, the proposed method does not have such problems.


- In addition, this work does not require manual labeling of generated images, but requires manually labeled real data.


- The learned annotator is student-agnostic


- A naive counterpart could be the GANinversion baseline. It is well-compared in the supplementary.


- The authors tackle a good scope range that has not been dealt with similar works, DatasetGAN and RepurposeGAN.


- Diverse experimental setups



<Cons>


- The mathematical derivation and formulation are primarily borrowed from [Zhao et al., ICLR 2021]. In this work, the synthetic dataset in [Zhao et al., ICLR 2021] is parameterized by GAN and the learnable annotator. This is on the borderline.


- While the authors argued that the proposed method significantly outperforms the semi-supervised competing methods, but the data regime is limited to low-data regimes. The effectiveness of this approach is unknown for the large semi-supervised data regimes (sufficient labeled data + very large unlabeled data).


- While the proposed approach has merits, its practicality is questionable in that the approach assumes the high fidelity pre-trained GAN. It would be very interesting to see how much robust the proposed method is with immaturely pre-trained GANs (i.e., low-quality GAN).


- The experiment settings are a bit biased to favorable settings to the proposed methods. Including the above concern, the experiment setups of DatasetGAN and RepurposeGAN are less considered. Are there specific reasons? Also, large-scale experiments and diverse balance setups could have been regarded.


<Questions>

- In the 1st paragraph of "Comparison to DatasetGAN", the authors mentioned that "FID cannot depict the disentanglement of generator features." What is the relationship of the proposed approach with disentanglement?


- The authors argued the non-necessity of labeling generated images as a benefit of the proposed method (including the one in "Comparison to DatasetGAN", Sec. 4). However, the proposed method requires the labeled real dataset as well. In order to validate the argument, the amount of the manual annotation cost (the number of labeled data) should be compared.



<Minor comments>


- The notation convention: In the paper, $\mathbf{\hat y}$ denotes manual annotation, while $\mathbf{y}$ denotes the predicted output. Conventionally, it it more often the other way around.


- $\mathcal{D}_l$ is abused for both a real labeled dataset and a labeled dataset of generated images, which may lead to confusion. It would be clearer to distinguish the notation.

- In Algorithm 1, the annotator and segmentor (called as student) are alternatingly optimized by each other. In this manner, the term student is misleading, since the role of the student is altered.

- Typos and grammars: There are many grammar errors, especially with articles, singular and plural forms.


**Summary Of The Paper:**

This work deals with the problem of training a part segmentation network by automatically synthesizing pairs of images and annotations. The authors propose a training method of an annotator given the well-pretrained generator model. The formulation is well-motivated and converged to the gradient matching loss.

This work extends [Zhao et al., ICLR 2021] by combining the idea of DatasetGAN and RepurposeGAN in an interesting way.
The proposed approach can favor the unlimited number of annotated data with only limited supervised real data + the pre-trained GAN.

The proposed method shows noticeable improvement over the other baselines and the competing methods in scarcely labeled data regimes (especially semi-supervised learning).


**Summary Of The Review:**

The submission has a few limitations, including the limited demonstration of its effectiveness in low-data regimes and grammatic errors. However, this work extends [Zhao et al., ICLR 2021] by combining the idea of DatasetGAN and RepurposeGAN in an interesting way. The merits of this work seem to outweigh the demerits.

===== After rebuttal =====
This reviewer stays at the same rating because the merits of this work seem to outweigh the demerits even after the discussion phase.
It would have been good to show the limitation and break-down point analyses in the submission.

---

> ### Author Response · Authors · 2021-11-20
> **Reply to Reviewer y2iP [2/2]**
>
> **Q4: The experiment settings are a bit biased to favorable settings to the proposed methods. Including the above concern, the experiment setups of DatasetGAN and RepurposeGAN are less considered. Are there specific reasons? Also, large-scale experiments and diverse balance setups could have been regarded.**
>
> We admit the experiment settings are a bit biased to low-data regime. In fact, this is also the experiment setups considered in DatasetGAN and RepurposeGAN. The reason is that the potential of this kind of methods on a large-scale data regime is still far from being fully revealed (as discussed above). Therefore, in this paper, we would like to focus more on the presentation of a low-data regime, where the results are very promising.
>
> **Q5: What is the relationship of the proposed approach with disentanglement?**
>
> Our work employs a very shallow network to convert generative features into annotations. Informally, we hypothesize the feasibility of using shallow networks is attributed to the disentangled representations learned by generative networks. Generally, disentanglement suggests that one factor of variation can be changed with other factors unchanged. This can be further understood as the semantic variation can be fit with simple models. To this end, one only needs a very shallow network (with perhaps low expressity) to convert the features into semantic annotations
>
> **Q6: However, the proposed method requires the labeled real dataset as well. In order to validate the argument, the amount of the manual annotation cost (the number of labeled data) should be compared.**
>
> Thank you for your questions. In the experimental comparison to DatasetGAN, we intend to make a fair comparison to DatasetGAN. Therefore, we train our annotator over *exactly the same* synthetic images used in DatasetGAN experiments. The number of labeled data is also the same as DatasetGAN: 30 for Cat-16, 16 for Face-34, 16 for Car-20. We have revised our paper to make the experimental setup and logic more clear.
>
> **Q7: Issues about notations, terms, typos and grammars.**
>
> Many thanks to the reviewer for the careful check of details. We have fixed these issues in our revised version.

---

> > ### Comment · Reviewer_y2iP · 2021-11-29
> > **Rely to the authors' response**
> >
> > Thank the authors for their responses.
> >
> > Although the authors did not well address my concerns and questions, after reading all the reviewers' comments and the authors' responses, this reviewer thinks that the benefits of this submission are a little bit outweigh the demerits of this work (details are below).
> > Thus, this reviewer stays at the same rating.
> >
> > For the authors' responses, while some of them are agreed, this reviewer has a few comments on them.
> >
> > - To the response to Q1: I didn't mean those methods and your work are the same, but what is different. The technical approach and derivation indeed share a similarity.
> >
> >
> > - To the response to Q2: My question was on Fig. 2, where the performance of the proposed method appears to be saturated, while the others having the same performance are not. This raises curiosity about what happens beyond this point. The authors failed to address this concern.
> >
> >
> > - To the response to Q3: The new evaluation looks good. The breaking point analysis should be reported in the paper, because this work has clear limitations. One of clear concerns I and Reviewer 318j shared was the limitation of this work, which would have been better to discuss more clearly in the initial submission.
> >
> >
> > - To the response to Q4: This reviewer disagrees with the authors who said, "we would like to focus more on the presentation of a low-data regime, where the results are very promising." What the authors said was clearly cherry-picking. As a scientific and engineering paper, a paper should report objective facts. Even though most of the advantage of the proposed method is reported in their playground (where the proposed method is favorable) as main reports, the authors are also strongly encouraged to provide the other sides, so that the readers can understand what the pros and cons, as well as the further research direction to go, are.
> > This point is a bit disappointing.

---

> > > ### Author Response · Authors · 2021-11-29
> > > **About large-scale setting**
> > >
> > > Thanks for your further feedback!
> > >
> > > We would like to kindly remind that our main focus is learning in very low-label scenarios, where collecting labels (pixel-wise annotations) are very expensive. We considered the same goal while designing our method rather than cherrypicking the best setting post-hoc. We also explicitly state that our method is verified for extremely limited labels throughout the submission and should not be penalized for any overclaiming.
> > >
> > > Additionally, we strongly agree with your point about providing objective evaluation. To provide a thorough analysis, we are currently running the methods in Fig.2 with more labeled data. We will include and discuss the complete analysis in the final version.

---

> > > > ### Comment · Reviewer_y2iP · 2021-11-29
> > > > **Response to the authors**
> > > >
> > > > Dear the authors,
> > > >
> > > > It is not penalized by the overclaiming but commented.
> > > >
> > > > Again, the authors argued that collecting labels is very expensive. However, there are either large-scale datasets for the part segmentation or complicated scenes that are hard to model by the recent generative model (as pointed out by Reviewer 318j).
> > > > In this regard, identifying the position of this work would be an important aspect in this work; where optimal trade-offs of the proposed approach are.
> > > >
> > > > My point was that this work would have been better valued by the other reviewers if being positioned better.
> > > >
> > > >
> > > > Nonetheless, the proposed method showed its benefits in low-data regimes well. One remaining concern by Reviewer 318j was proper baselines. Despite this, in my personal opinion, this work is a reasonable paper that can be accepted.
> > > > However, I would not strongly champion and respect the other reviewers' opinions.

---

> ### Author Response · Authors · 2021-11-20
> **Reply to Reviewer y2iP [1/2]**
>
> **Q1: The mathematical derivation and formulation are primarily borrowed from [Zhao et al., ICLR 2021]. In this work, the synthetic dataset in [Zhao et al., ICLR 2021] is parameterized by GAN and the learnable annotator.**
>
> Thanks for your comments. It is very interesting to understand our method as “parameterizing synthetic data in [Zhao et al., ICLR 2021] with GAN and the learnable annotator”. We would like to state the connection and difference between our work and Dataset Condensation (DC) (Zhao et al., 2021) as follows and a similar discussion is updated in the related work section.
>
> “Synthetic dataset” refers to different things in our work and DC. Our work requires photo-realistic synthetic images, whereas DC does not care about the realness of the synthetic images (a.k.a. condensed images) and the condensed images are typically not realistic at all.
>
> Our goal is also different from DC. While our method learns to label photo-realistic synthetic images for training segmentation networks, Zhao et al., (2021) learn to synthesize a small set of images to condense a large set of real images for training image classification networks.
>
> **Q2: The effectiveness of this approach is unknown for the large semi-supervised data regimes (sufficient labeled data + very large unlabeled data).**
>
> Thank you for this interesting point. The experiments on CelebA in Figure 2 (left) presents results for the suggested setting for 150 and 1500 labeled images with 28,000 unlabeled images. As expected, the gains are smaller in the large labeled data regime, as the results are already close to the supervised learning baseline. However, our method still gains comparable and even better performance than other SSL segmentation methods (e.g. AdvSSL, GCT).
>
> **Q3: While the proposed approach has merits, its practicality is questionable in that the approach assumes the high fidelity pre-trained GAN. It would be very interesting to see how much robust the proposed method is with immaturely pre-trained GANs (i.e., low-quality GAN).**
>
> As suggested, we provide a new experimental comparison when GANs of different quality are used. The results are as follows.
>
> |  | CelebA | CelebA | CelebA | CelebA | Aeroplane |  Aeroplane | Aeroplane | Aeroplane |
> | :--: | :---: | :---: | :---: | :---: | :------: | :-----: | :--------: | :------: |
> | FID | 10.21 | 6.80 | 5.12 | 4.29 | 11.74 | 7.78 | 6.25 | 5.04 |
> | FG-mIoU | 73.92 | 76.61 | 77.48 | 78.07 | 21.69 | 36.01 | 38.81 | 41.21 |
>
> The quality of pre-trained GANs plays an important role in our method. First, our segmentation network is directly trained only on the generated images (Alg.1 Line 9~11). Second, we employ a very shallow network to convert generative features into annotations, which suggests that the quality of generative features significantly affect the quality of annotations. Therefore, if the GAN generates low-quality images or maintains low-quality generative features, the segmentation performance inevitably degrades.

---

### Official Review · Reviewer_Z2yc · 2021-11-02

**Correctness:** 3
**Technical Novelty And Significance:** 3
**Empirical Novelty And Significance:** 3
**Recommendation:** 8
**Confidence:** 3

**Main Review:**

This paper has the following Strengths.

- The problem covered by the paper is important, and a good direction is proposed based on clear motivation.
- Experiments were sufficiently conducted to verify the effectiveness of the proposed method.
- The paper is written in an easy-to-understand manner through detailed formulas and figures.

However, the following concerns exist.

1. Comparison with the end-to-end gradient-based meta-learning approach
    - In section 3.2, it is stated that the nested-loop optimization problem can be solved through existing end-to-end gradient-based meta-learning approaches. And the authors claim that those methods are complex and expensive, so they suggest an efficient gradient matching approach. However, I cannot find any experiment to show how efficient and effective the proposed method is compared to them, and this is my biggest concern.
2. Images in batch to get gradient matching loss
    - According to Algorithm 1, the images used to compute each gradient appear to be identical. (The indices of $x_i$ for $B_l$ and $B_g$ are the same.) However, in Figure 1, the two images for each gradient are different, causing confusion.
3. Pascal-{Horse, Aeroplane} dataset
    - For this paper, images of Horse and Aeroplane were selected from the Pascal Part segmentation dataset for the experiment. Is there any particular reason for choosing these two classes (among 20 classes)?


**Summary Of The Paper:**

This paper proposes a method to solve the problem that generally requires large-scale labeled datasets to train deep learning models. The proposed method is designed to solve the nested-loop optimization problem (with an annotator that generates a label and a student network that predicts a label) based on a pre-trained GAN that generates images. In particular, the authors solve this problem through a gradient matching approach, and they claim it is much more efficient than the existing end-to-end gradient-based meta-learning approach. This method was applied to the semi-supervised part segmentation task, and its effectiveness was verified through various experiments.


**Summary Of The Review:**

This paper proposes an intuitive method based on clear motivation. In addition, sufficient experiments have been conducted to prove it, and the paper is written in an easy-to-understand manner. However, one big concern (#1 above) remains, so I give my initial rating as 6.

**[Comments After Author Response]** I appreciate the efforts of the authors for the detailed rebuttal. The authors have well addressed my concerns (other reviewers' as well), so I raise my rating to 8. I strongly recommend that the authors will reflect those contents in the final version.

---

> ### Author Response · Authors · 2021-11-20
> **Reply to Reviewer Z2yc**
>
> **Q1: Comparison with the end-to-end gradient-based meta-learning approach.**
>
> As suggested, we report results for 1-step/2-step inner-loop MAML. We limit the number of steps to 2 while training MAML due to the high memory requirements. The results are also attached as follows.
>
> | | G | S | Cat-16 | Face-34 | Cat-20 |
> | :---: | :---: | :---: | :---: | :---: | :---: |
> | Ours-GM | StyleGAN | DeepLab | 33.89±0.43 | 52.58±0.61 | 63.55±2.25 |
> | Ours-GM | StyleGAN | UNet | 32.64±0.74 | 53.69±0.54 | 60.45±2.42 |
> | Ours-GM | StyleGAN2 | DeepLab | 33.56±0.17 | 55.10±0.39 | 61.21±2.07 |
> | Ours-GM | StyleGAN2 | UNet | 31.90±0.75 | 53.58±0.45 | 58.30±2.64 |
> | Ours-MAML (1-step) | StyleGAN | DeepLab | 32.49±0.43 | 35.24±0.20 | 54.74±2.67 |
> | Ours-MAML (1-step) | StyleGAN | UNet | 16.70±0.43 | 23.21±0.07 | 29.26±0.72 |
> | Ours-MAML (1-step) | StyleGAN2 | DeepLab | 30.30±0.53 | 33.82±0.24 | 54.42±3.11 |
> | Ours-MAML (1-step) | StyleGAN2 | UNet | 22.78±0.65 |32.64±0.18 | 26.59±0.60 |
> | Ours-MAML (2-step) | StyleGAN | DeepLab | 32.98±0.50 | 39.73±0.13 | 58.22±2.17  |
> | Ours-MAML (2-step) | StyleGAN | UNet | 14.20±0.26 | --- | --- |
> | Ours-MAML (2-step) | StyleGAN2 | DeepLab | 29.66±0.69 | 39.04±0.23 | 56.47±2.32 |
> | Ours-MAML (2-step) | StyleGAN2 | UNet | 19.37±0.35 | --- |  --- |
>
> Note: The “---” indicates that we fail to obtain reasonable results
>
> Our analysis and conclusion are:
>
> - Gradient matching generally leads to higher performance and obtains more consistent results across different datasets and different segmentation network architectures.
>
> - In practice, we observe MAML is more sensitive to hyperparameters such as inner-loop learning rate, whereas our method nearly requires neither dataset-specific nor network-architecture-specific hyperparameters.
>
> In terms of efficiency, our method only gains a bit of memory savings compared to 1-step MAML. However, as suggested by the above results, MAML fails to match the performance of gradient matching with 2-step inner loop and probably requires unrolling more inner-loop steps to achieve it, which is apparently much more inefficient than gradient matching.
>
> **Q2: Images in batch to get gradient matching loss.**
>
> Thanks for your kind suggestion. We have fixed this issue in our revised version.
>
> **Q3: Pascal-{Horse, Aeroplane} dataset.**
>
> We chose the horse and aeroplane mainly out of two concerns. First, we select the class for which pre-trained GAN models are easily obtained. For example, the pre-trained StyleGAN2 model on LSUN horse is released by Karras et al., (2020). Second, these categories are common in various computer vision datasets such as MS COCO (Lin et al., 2014), ImageNet (Deng et al., 2009) and there are some works (Wang and Alan, 2015; Zhao et al., 2019)  focusing on parsing animal and vehicle parts.
>
> [1] Lin, Tsung-Yi, Michael Maire, Serge Belongie, James Hays, Pietro Perona, Deva Ramanan, Piotr Dollár, and C. Lawrence Zitnick. Microsoft coco: Common objects in context. In ECCV, 2014.
>
> [2] Deng, Jia, Wei Dong, Richard Socher, Li-Jia Li, Kai Li, and Li Fei-Fei. Imagenet: A large-scale hierarchical image database. In CVPR, 2009.
>
> [3] Wang, Jianyu, and Alan L. Yuille. Semantic part segmentation using compositional model combining shape and appearance. In CVPR, 2015.
>
> [4] Zhao, Yifan, Jia Li, Yu Zhang, Yafei Song, and Yonghong Tian. Ordinal multi-task part segmentation with recurrent prior generation. TPAMI, 2019.

---

> > ### Comment · Reviewer_Z2yc · 2021-11-30
> > **Comments After Author Response**
> >
> > Thanks for the detailed rebuttal. All of my concerns were addressed well, and I changed my rating accordingly. I strongly recommend that the authors will reflect those contents in the final version.

---

### Official Review · Reviewer_318j · 2021-11-04

**Correctness:** 3
**Technical Novelty And Significance:** 3
**Empirical Novelty And Significance:** 3
**Recommendation:** 6
**Confidence:** 4

**Main Review:**

Strengths:
- The paper makes a valuable improvement over the DatasetGAN-like approaches by removing the need for annotating synthetic samples.
- The motivations and the proposed solution is clearly explained in the paper.
- The approach also appears to be an effective alternative for traditional semi-supervised learning. The experimental comparison to semi-supervised methods mostly appear to be convincing enough.
- The paper also reports promising results in comparison to DatasetGAN, though the results are a bit mixed in terms of the performance.

Weaknesses:
- SemanticGAN's relevance is not clearly discussed, despite the experimental comparisons made in Section 4. In particular, it should be clarified in intro/related work, whether this work proposes an alternative approach to what SemanticGAN does, or whether they differ in terms of the data/annotation requirements. Please clarify.
- The paper seems to have missed a closely related work:  Gradient Matching Generative Networks for Zero-Shot Learning (CVPR 2019). This CVPR’19 paper appears to introduce the “gradient matching loss" (the same core formulation, and possibly its name as well) for training conditional generative models, as an efficient approximation to inner loop optimization for meta-learning. It trains synthetic train data-generating conditional generative models, such that when a classification model trained over the synthetic samples the resulting model’s loss is low on limited labelled samples, which is clearly closely. While the overall differences are significant enough (eg training conditional generator vs annotator only, segmentation vs zero-shot classification), the core idea, the use of Gradient Matching Loss and the overall goal of leveraging generative models for labelled data generation via meta-learning principles appear to be closely related, therefore, need to be discussed in the paper.
- The experimental comparison to DatasetGAN is not fully clear (to me). DatasetGAN uses the annotated synthetic samples. Which labelled examples do you use for the proposed approach in these comparison experiments? It would have been valuable to do this in two ways: (i) train over the synthetic images used in DatasetGAN experiments, and (ii) train over alternative real training images.
- The paper does not discuss whether it can be used on complex scenes, given that it is typically much more difficult to obtain unsupervised generative models of complex scenes.
- It would have been very valuable to see an evaluation that shows the detailed analysis on the annotation cost vs final segmentation accuracy (in mIoU), when real images are labelled & used with the proposed approach, in comparison to labelling synthetic images to be used in DatasetGAN. Currently it is unclear how many more real images need to be annotated to match the DatasetGAN's sample generation performance, given that DatasetGAN performs better in several cases (Table 1).
- Fig 2: what happens if you train fine-tuned the model only on real annotated images, using the same training procedure, starting with pre-trained backbone, with optimized hyper-parameters?
- How would it perform if you were to train the segmentation model solely on the available real samples, apply it to synthetic images for pseudo-labeling the pixels? This baseline also seems to be missing.
- Please clarify at which stage(s) the synthetically generated images are being used in the semi-supervised experiments (Figure 2). Do you use the synthetic images as the unsupervised image set?
- How would it perform if you were to actually do the inner loop optimization? While the computational advantages are clear, the difference is not empirically evaluated. In particular, how would the approach work with a 1st-order / kth-order MAML-based loss?


**Summary Of The Paper:**

The ultimate goal of the paper is to train an unsupervised generative model, in a semi-supervised manner, as a synthetic training example generator for training segmentation models. Prior works (eg. DatasetGAN) show that one can label a few synthetic images from a pre-trained generative model, and train a segmentation model (called annotation model) over the intermediate representation produced by the generative model. Then, every time a sample is taken from the generative model, the corresponding labels can be produced by applying the annotation model to the corresponding intermediate representations. This provides a valuable way to produce synthetic labelled training examples. However, (some of) these existing approaches require annotating (a limited set of) generated samples for training the annotation model. This is problematic as one needs to re-label such images every time the generative model changes.

This paper aims to address this problem by learning the annotation model over annotated real images instead of annotated synthetic images. To this end, the paper proposes a meta-learning approach. The main idea is to learn the annotation model such that when a (student) segmentation model is trained over the generative model outputs and their labels given by the annotation model, the resulting student model’s loss shall be low on real annotated images. This meta-learning idea is implemented as an approximation by using the Gradient Matching Loss.


**Summary Of The Review:**

The paper proposes a reasonable and interesting meta-learning approach towards leveraging (pre-trained) generative models for constructing synthetic labelled training sample sets. The overall approach is interesting, the results are promising yet the paper (in text and in experimental analysis) seems to have various weakness, as discussed above.

While I do *not* consider that the existence of all suggested experimental evaluations is a pre-condition for acceptance, the current experimental analysis seem to be underwhelming in several aspects. In addition, all the textual weaknesses & ambiguities definitely need to be addressed/clarified in the comments/revision.

— Post revision/rebuttal update —
I have gone over all the reviews, responses and the updates in the paper. The revision and responses address my concerns. The text and experimental evaluation have been improved on several aspects, though there is still room for improvement in experimental analysis. Overall, based on all positive improvements, I increase my recommendation score from 5 to 6.

---

> ### Author Response · Authors · 2021-11-20
> **Reply to Reviewer 318j [3/3]**
>
> **Q7: ... pseudo-labeling ... baseline also seems to be missing.**
>
> We have included the pseudo-labeling results in the updated submissions.
>
> The experimental setting is as follows. First, we train a segmentation model on the labeled examples. Second, each randomly generated image is fed into the trained segmentation model to obtain the prediction as the pseudo-label which trains the annotator in a supervised manner. Third, the trained joint annotator and generator are used for generating automatically labeled data and training a downstream segmentation network. Results are as follows.
>
> | Method | G | S | Cat-16 | Face-34 | Car-20 |
> |:---:|:---:|:---:|:---:|:---:|:---:|
> | Ours | StyleGAN2 | DeepLab | 33.56±0.17 | 55.10±0.39 | 61.21±2.07 |
> | Ours | StyleGAN2 | U-Net   | 31.90±0.75 | 53.58±0.45 | 58.30±2.64 |
> | Pseudo-labeling baseline | StyleGAN2 | DeepLab | 20.21±0.65 | 42.15±0.68 | 12.37±0.91 |
> | Pseudo-labeling baseline | StyleGAN2 | U-Net   | 16.72±0.79 | 43.26±1.20 | 13.48±0.48 |
>
> We think it is not surprising that our method outperforms this simple baseline with a large margin. The insight of our method is to distill knowledge from a generative model to a discriminative model, whereas this pseudo-labeling baseline is more like the other way around, distilling knowledge from a discriminative model to a generative model. The highly interpretable generator features make our method able to learn annotations from only a few labeled data. In contrast, the discriminative model can hardly learn to acquire valuable knowledge from extremely limited labeled data.
>
>
> **Q8: Please clarify at which stage(s) the synthetically generated images are being used in the semi-supervised experiments (Figure 2). Do you use the synthetic images as the unsupervised image set?**
>
> In Fig.2, given a pre-trained GAN, our method only involves one stage which iteratively trains the annotator and the segmentation network. As algorithm 1 (line 4-5; line 9-11) indicates, the synthetically generated images are used for both the annotator learning step and the segmentation network learning step.
>
> Other SSL methods do not use any “synthetically generated images” and directly use the real unlabeled images as the “unsupervised image set”, which is the same training data for GAN.
>
>
> **Q9: How would it perform if you were to actually do the inner loop optimization? ... how would the approach work with a 1st-order / kth-order MAML-based loss?**
>
> As suggested, we report results for 1-step/2-step inner-loop MAML. We limit the number of steps to 2 while training MAML due to the high memory requirements. The results are also attached as follows.
>
> | | G | S | Cat-16 | Face-34 | Cat-20 |
> | :---: | :---: | :---: | :---: | :---: | :---: |
> | Ours-GM | StyleGAN | DeepLab | 33.89±0.43 | 52.58±0.61 | 63.55±2.25 |
> | Ours-GM | StyleGAN | UNet | 32.64±0.74 | 53.69±0.54 | 60.45±2.42 |
> | Ours-GM | StyleGAN2 | DeepLab | 33.56±0.17 | 55.10±0.39 | 61.21±2.07 |
> | Ours-GM | StyleGAN2 | UNet | 31.90±0.75 | 53.58±0.45 | 58.30±2.64 |
> | Ours-MAML (1-step) | StyleGAN | DeepLab | 32.49±0.43 | 35.24±0.20 | 54.74±2.67 |
> | Ours-MAML (1-step) | StyleGAN | UNet | 16.70±0.43 | 23.21±0.07 | 29.26±0.72 |
> | Ours-MAML (1-step) | StyleGAN2 | DeepLab | 30.30±0.53 | 33.82±0.24 | 54.42±3.11 |
> | Ours-MAML (1-step) | StyleGAN2 | UNet | 22.78±0.65 |32.64±0.18 | 26.59±0.60 |
> | Ours-MAML (2-step) | StyleGAN | DeepLab | 32.98±0.50 | 39.73±0.13 | 58.22±2.17  |
> | Ours-MAML (2-step) | StyleGAN | UNet | 14.20±0.26 | --- | --- |
> | Ours-MAML (2-step) | StyleGAN2 | DeepLab | 29.66±0.69 | 39.04±0.23 | 56.47±2.32 |
> | Ours-MAML (2-step) | StyleGAN2 | UNet | 19.37±0.35 | --- |  --- |
>
> Note: The “---” indicates that we fail to obtain reasonable results
>
>
> Our analysis and conclusion are:
>
> - Gradient matching generally leads to higher performance and obtains more consistent results across different datasets and different segmentation network architectures.
>
> - In practice, we observe MAML is more sensitive to hyperparameters such as inner-loop learning rate, whereas our method nearly requires neither dataset-specific nor network-architecture-specific hyperparameters.

---

> > ### Comment · Reviewer_318j · 2021-11-29
> > **Post discussion**
> >
> > Thanks for the new experimental results and clarifications/improvements in the paper’s text. I have updated my review accordingly.

---

> ### Author Response · Authors · 2021-11-20
> **Reply to Reviewer 318j [2/3]**
>
> **Q5: … an evaluation that shows the detailed analysis on the annotation cost vs final segmentation accuracy… Currently it is unclear how many more real images need to be annotated to match the DatasetGAN...**
>
> We already present an evaluation of the segmentation performance with respect to the annotation cost (indicated by number of labeled images) when real images are used in Fig.2.
>
> When synthetic images are used, we limit ourselves to the annotated images released by the DatasetGAN, as we are unable to collect more annotated images. Based on this, we conducted an experiment that compares our method to DatasetGAN for a varying number of annotated synthetic images in Car 20, where 33 such images are available from DatasetGAN. The results suggest that with an increased number of labeled data, our method approaches the performance of DatasetGAN.
>
> | G | S | DatasetGAN | Ours | Ours | Ours |
> | :---: | :---: | :---: | :---: | :---: | :---: |
> |  |  | **# labels: 16** | **# labels: 16** | **# labels: 25** | **# labels: 33** |
> | StyleGAN | DeepLab | 67.53±2.58 | 63.55±2.25 | 64.68±2.53 | 64.88±2.52 |
> | StyleGAN | U-Net   | 66.27±2.75 | 60.45±2.42 | 63.09±3.49 | 65.37±2.78 |
>
>
> **Q6: Fig 2: what happens if you train fine-tuned the model only on real annotated images, using the same training procedure, starting with pre-trained backbone, with optimized hyper-parameters?**
>
> We are not sure that we fully understand your question and motivation but we assume that three new baselines are required under the setting of Fig. 2:
> - SL: Supervised learning.
> - Ours + f.t.: First train a segmentation network using our algorithm and then finetune it on labeled data.
> - Pretrain + Ours: First train a segmentation network on labeled data and then finetune it using our algorithm.
>
> The comparison of our method to these additional baselines is presented as below
>
> - CelebA
>
> |  | SL | Ours | Ours+f.t. | Pretrain+Ours |
> |:---:|:---:|:---:|:---:|:---:|
> 2 | 27.89 | 56.55 | 33.58 | 57.88 |
> 10 | 63.53 | 74.41 | 64.65 | 72.48 |
> 30 | 72.20 | 77.23 | 72.93 | 76.40 |
> 150 | 78.44 | 77.96 | 78.53 | 76.83 |
> 1500 | 82.12 | 78.07 | 82.26 | $\textcolor{grey}{82.08}$ |
>
> - Horse
>
> |  | SL | Ours | Ours+f.t. | Pretrain+Ours |
> |:---:|:---:|:---:|:---:|:---:|
> | 2 | 9.55 | 37.25 | 12.89 | 23.28 |
> | 10 | 21.75 | 48.59 | 22.54 | 39.31 |
> | 30 | 32.95 | 54.29 | 33.50 | $\textcolor{grey}{33.13}$ |
> | 100 | 41.33 | 56.16 | 42.26 | $\textcolor{grey}{41.52}$ |
> | 180 | 46.85 | 56.83 | 46.88 | $\textcolor{grey}{47.18}$ |
>
> - Aeroplane
>
> |  | SL | Ours | Ours+f.t. | Pretrain+Ours |
> |:---:|:---:|:---:|:---:|:---:|
> | 2 | 2.15 | 29.01 | 7.65 | 27.14 |
> | 10 | 16.97 | 34.29 | 17.88 | 24.27 |
> | 30 | 24.29 | 38.66 | 26.94 | $\textcolor{grey}{24.42}$ |
> | 100 | 33.01 | 42.94 | 32.28 | $\textcolor{grey}{33.03}$ |
> | 180 | 36.92 | 41.21 | 37.52 | $\textcolor{grey}{36.97}$ |
>
> $\textcolor{grey}{grey~number}$: the checkpoints achieving the best validation performance is the initial checkpoint (pretrained one), which suggests that finetuning using our algorithm does not really work.
>
>
> It can be seen from the above results that:
>
> - “Ours+f.t.” mostly outperforms “SL”, indicating that our algorithm provides a better initialized point for the segmentation network.
> - The performance of “Ours+f.t.”  is more related to “SL” due to the domination of labeled data during the finetuning.
> - “Pretrain+Ours” mostly harms the performance. Note that we target the annotator learning (Eq.6) at producing labels that result in a learned segmentation network performing well on the labeled data. In this way, the automatically labeled data is like the “training set” while the manually labeled data is like the “validation set”. If the segmentation network is pretrained on the labeled data, it violates the rule that the training set should not intersect with the validation set. Therefore, it might be a bit meaningless to pre-train the segmentation network on the labeled data.

---

> ### Author Response · Authors · 2021-11-20
> **Reply to Reviewer 318j [1/3]**
>
> **Q1: SemanticGAN's relevance is not clearly discussed ...**
>
> Thanks for the suggestion. We now discuss SemanticGAN at the end of the third paragraph in the introduction
> > ... Another way to address this issue is to align the joint distribution of generated images and labels with that of real ones using adversarial learning as in SemanticGAN (Li et al., 2021). However, adversarial learning is notoriously unstable (Brock et al., 2018) and requires sufficient data to prevent discriminator from overfitting (Karras et al., 2020a).
>
> as well as at the end of the second paragraph in the related work
> > ... Particularly, unlike SemanticGAN (Li et al., 2021) that learns annotator and data generator jointly, our method presumes a fixed pre-trained GAN and learns annotator only, which circumvents the complication of joint learning.
>
> **Q2: The paper seems to have missed a closely related work: Gradient Matching Generative Networks for Zero-Shot Learning (CVPR 2019)**
>
> Thanks for pointing out the missing related work. Now we include and discuss this reference in the “Gradient matching” paragraph of the related work section.
>
> > ... Although the general principle and gradient matching loss function in our work resemble those in the above works, we are motivated to solve a totally different problem. In particular, Gradient Matching Network (GMN) (Sariyildiz & Cinbis, 2019) employs gradient matching to train a conditional generative model towards application on zero-shot classification ...
>
> Thanks for your positive statement, “the overall differences are significant enough ...”. We would like to highlight that the existence of this paper does not undermine our contribution including, as summarized at the end of the introduction of the revised paper, (i) a method that mitigates the limitations of DatasetGAN-like approach, and (ii) empirical results that show the promise of using generative models, none of which is covered by this CVPR 2019 paper.
>
>
> **Q3: The experimental comparison to DatasetGAN … (could be done) in two ways: (i) train over the synthetic images ..., and (ii) train over alternative real training images.**
>
> For a fair comparison to DatasetGAN, we train our annotator over exactly the same synthetic images used in DatasetGAN experiments.
>
> As suggested, our method can also be trained over alternative real training images. The results of this comparison are as follows. In particular, DatasetGAN evaluates its performance on CelebA-test (8 classes) using 16 annotated synthetic images, which is shown in the first row. We also evaluate the performance of our method under this setting using (i) exactly the same synthetic images as DatasetGAN and (ii) the same number of real images taken from CelebA-train.
>
> Our method outperforms DatasetGAN and the performance can be further improved by replacing labeled synthetic images with labeled real images. This suggests that the capability of using real labeled data exhibits the superiority of our method over DatasetGAN.
>
> |        | Labeled data | CelebA-test@512x512 |
> | :---: | :---:                 | :-----: |
> | DatasetGAN | Synthetic | 70.01 |
> | Ours  | Synthetic | 72.55 |
> | Ours  | Real        | 79.25|
>
> Note: experiments are done using StyleGAN as generator and Deeplabv3 as segmentation network at 512x512 resolution.
>
> **Q4: The paper does not discuss whether it can be used on complex scenes, given that it is typically much more difficult to obtain unsupervised generative models of complex scenes.**
>
> We agree with you that it is typically much more difficult to obtain unsupervised generative models of complex scenes. We added discussion about this point in the introduction and the conclusion section.
>
> - Introduction:
> > ... Furthermore, considering that state-of-the-art unconditional GANs only produce appealing results on single-class images but struggle to model complex scenes, ...
>
> - Conclusion:
> > ... The effectiveness of our method is validated on a variety of single-class datasets including well-aligned images, e.g. CelebA and Face-34, as well as images in the wild--part of them containing cluttered backgrounds-- e.g. Pascal-Horse, Pascal-Aeroplane, Car-20, and Cat-16. In terms of more complex scenes, discussion and investigation are further needed. With the rapid progress of generative modeling, it is promising to use more powerful generative models ...
>
> The capability to handle complex scenes may be hopefully improved in the future with the rapid progress of generative modeling. In principle, our method is also possible to the issue of complex scenes.
>
> An alternative to address the issue of complex scenes would be using pre-trained object detectors to crop object instances in complex scenes and use them to train generative models. In this case, the object detector would require bounding box annotations which are much cheaper than pixel labels.

---

### Author Response · Authors · 2021-11-21
**Revision Summary**

We thank all the reviewers for their valuable feedback! We have made the following revision to our paper to address the reviewers’ concerns:

- Clarifying the difference of our method with SemanticGAN in the introduction & related work. ($\textcolor{red}{@Reviewer~318j}$)

- Adding a summary of our contribution in the introduction. ($\textcolor{orange}{@Reviewer~muji}$)

- Adding a discussion about a missing reference in the related work. ($\textcolor{red}{@Reviewer~318j}$)

- Adding a discussion about the difficulty of dealing with complex scenes in the introduction and conclusion. ($\textcolor{red}{@Reviewer~318j}$)

- Clarifying the number of labeled images in Table 1. ($\textcolor{orange}{@Reviewer~muji}$)

- Changing paragraph headings in Sec.4.2. ($\textcolor{orange}{@Reviewer~muji}$)

- Adding the performance of supervised learning baseline to Fig. 2. ($\textcolor{red}{@Reviewer~318j}$)

- Fixing issues of language, notation, figures, tables, typos, the origin of reported results, etc. ($\textcolor{green}{@Reviewer~Z2yc}$; $\textcolor{blue}{@Reviewer\ y2iP}$; $\textcolor{orange}{@Reviewer\ muji}$)

Furthermore, as suggested by the reviewers, we also enrich our experimental analysis and attached the additional results to the appendix:

- Comparison of our method to a “pseudo labeling” baseline: Sec.C.1 Pseudo-labeling baseline. ($\textcolor{red}{@Reviewer~318j}$)

- Comparison between MAML and gradient matching: Sec.C.1 Our method with MAML ($\textcolor{red}{@Reviewer\ 318j}$; $\textcolor{green}{@Reviewer\ Z2yc}$)

- Comparison of our method when real images versus synthetic images are used as labeled data: Sec.C.5 Real images v.s. synthetic images as labeled data. ($\textcolor{red}{@Reviewer~318j}$)

- Performance of our method on Car-20 when more labeled data is used: Sec.C.5 Our method with more labeled data on Car-20. ($\textcolor{red}{@Reviewer~318j}$)

- Performance of our method when using immature pre-trained GAN: Sec.C.4. ($\textcolor{blue}{@Reviewer~y2iP}$)

Finally, we would like to explain the update of the performance of other SSL methods in Fig.2. During the rebuttal period, we find the old results of other SSL methods have the following issues

- On CelebA, the old results are copied from the SemanticGAN (Li et al., 2021) paper.

- On Pascal-Horse and Pascal-Aeroplane, the old results are obtained by ourselves using implementations from [PixelSSL](https://github.com/ZHKKKe/PixelSSL) and DeepLabv2 since PixelSSL does not support DeepLabv3.

We realize this may lead to an unfair comparison since our work uses DeepLabv3. Now we have added the support of DeepLabv3 based on PixelSSL, rerun experiments, and updated the results of other SSL methods (MT, GCT, and AdvSSL) on CelebA, Pascal-Horse, and Pascal-Aeroplane. The implementation details about how we obtain these results are also updated in the appendix. According to the new results, Our method still outperforms other SSL methods when labeled data is extremely limited. Therefore, our experimental conclusions are unchanged.

---

### Decision · Program_Chairs · 2022-01-20

**Decision:**

Accept (Poster)

**Comment:**

This work deals with training generators of aligned pairs of images and segmentation maps. It is based on the recent DatasetGAN approach, which generates images and maps, but requires human annotations on a handful of generated images. This paper is addressing this problem by learning the annotation model over annotated real images instead of generated ones. To this end, the paper proposes a meta-learning that uses the Gradient Matching Loss.

Overall, the rebuttal provides valuable insight and many issues raised by reviewers have been convincingly answered by the authors.
On the whole, the reviewers converged positively, the novelty and the interest of the proposal stand out clearly, and this despite the lack of very convincing experiments, at least before the rebuttal. Authors are strongly encouraged to take all comments into account for their final version.